# Extreme Concentric Gravity Waves Observed in the Mesosphere and Thermosphere Regions over Southern Brazil Associated with Fast-Moving Severe Thunderstorms

Qinzeng Li[1], Jiyao Xu[1], Yajun Zhu[1], Cristiano M. Wrasse[2], José V. Bageston[3], Wei Yuan[1], Xiao Liu[4], Weijun Liu[1], Ying Wen[5], Hui Li[1], and Zhengkuan Liu[1]

1. State Key Laboratory of Solar Activity and Space Weather, National Space Science Center, Chinese Academy of Sciences, Beijing, 100190, China
2. National Institute for Space Research, Space Weather Division, São José dos Campos, SP, Brazil
3. National Institute for Space Research, Southern Space Coordination, Santa Maria, RS, Brazil
4. School of Mathematics and Information Science, Henan Normal University, Xinxiang, 453007, China
5. College of Aviation Meteorology, Civil Aviation Flight University of China, Guanghan, 618307, China

Correspondence: Jiyao Xu (jyxu@swl.ac.cn) and Yajun Zhu (y.zhu@swl.ac.cn)

**Abstract**

Three groups of intense concentric gravity waves (CGWs) lasting over 10 hours were observed by an airglow imager at the Southern Space Observatory (SSO) in São Martinho da Serra (29.44°S, 53.82°W) in southern Brazil on 17-18 September 2023. These CGW events were simultaneously captured by spaceborne instruments, including the Atmospheric Infrared Sounder (AIRS) aboard Aqua, the Visible Infrared Imaging Radiometer Suite (VIIRS) onboard Suomi NPP, and the Sounding of the Atmosphere using Broadband Emission Radiometry (SABER) instrument operating on the Thermosphere-Ionosphere-Mesosphere Energetics and Dynamics (TIMED) satellite. The CGW caused significant airglow radiation perturbations exceeding 24% and the distance of the wave center movement exceeded 400 km. These CGW events were caused by fast-moving deep convection observed by Geostationary Operational Environmental Satellite-16 (GOES-16). The weaker background wind field during the spring season transition provides the necessary conditions for CGWs to propagate from the lower atmosphere to the mesopause region. The 630 nm emission images were significantly contaminated by specific OH emission bands. The same CGW event was observed propagating from the OH airglow layer (~87 km) to the thermospheric OI 630.0 nm airglow layer (~250 km). The asymmetric propagation of CGWs in the thermosphere may be due to the vertical wavelength changes caused by the Doppler-shifting effect of the background wind field. This multi-layer ground-based and satellite joint detection of CGWs offers an excellent perspective for examining the coupling of various atmospheric layers.

# 1. Introduction

Atmospheric gravity waves (AGWs) are disturbances in the atmosphere caused by various sources, such as convection (Fovell et al. 1992; Piani et al. 2000; Heale et al., 2021; Franco-Diaz et al., 2024), front/jet stream (Fritts and Nastrom, 1992; Plougonven and Zhang 2014; Dalin et al., 2016; Wrasse et al., 2024), wind shear (Fritts, 1982; Pramitha et al., 2015), orography forcing (Nastrom and Fritts, 1992; Wright et al., 2017; Liu et al., 2019; Heale et al., 2020; Geldenhuys et al., 2021; Inchin et al., 2024), and air–sea interaction (Li et al., 2024). AGWs are generated when strong updrafts and downdrafts displace the stable stratification of the atmosphere. As AGWs propagate vertically from the lower atmosphere, their amplitude grows markedly owing to reduced density. When they reach mesosphere–lower thermosphere (MLT) altitudes, they become unstable and break, dissipating momentum and energy into the surrounding atmosphere (Cao and Liu, 2016; Ern et al., 2022). This energy deposition makes AGWs crucial drivers of the momentum and energy budgets in the MLT region, fundamentally governing the general circulation, thermal structure, chemical composition distribution, and transport regimes (Fritts and Alexander, 2003; Plane et al., 2023).

Among the many sources of AGWs, convective sources are particularly significant (Alexander and Holton, 2004). They can generate concentric gravity waves (CGWs), the source location of which can be readily determined by the center position. The backward ray tracing method, employed for source location determination, can also be applied to circular gravity wave patterns (Ern et al. 2022).

This enables point-to-point studies of their propagation characteristics. The
release of latent heat in deep convection acts as a forcing mechanism (Lane et al.,
2001), creating CGWs that can propagate upward into the middle and upper
atmosphere.
All-sky airglow imagers provide a large field of view and high-resolution
observations, making them particularly suitable for observing short-period AGWs
in the mesosphere and thermosphere. Through the observational data from airglow
imagers, researchers can analyze the propagation characteristics of AGWs,
including parameters such as horizontal wavelengths, observed periods,
horizontal phase velocities and momentum fluxes (Swenson and Liu, 1998).
Although the observation of AGWs by airglow imagers has been widely
documented in previous studies (Dalin et al., 2024; Nyassor et al., 2021, 2022;
Suzuki et al., 2007a; Vadas et al., 2012; Vargas et al., 2021; Wüst et al., 2019; Xu
et al., 2015; Yue et al., 2009), dual-layer airglow observations, which involve
observing airglow emissions from a hydroxyl radical (OH) layer (~87 km) in the
mesosphere and an atomic oxygen emission layer at 630 nm (OI 630.0 nm) (~250
km) in the thermosphere, offer a unique opportunity to simultaneously investigate
CGWs in both the mesosphere and thermosphere. This configuration enables
comprehensive studies of gravity wave vertical propagation and their role in
vertical atmospheric coupling. However, due to past limitations in observational
capabilities, simultaneous detection of CGWs across both the OH and OI 630.0 nm
layers was rare.

In this study, we observed multiple strong CGW events using airglow measurements in southern Brazil on 17-18 September 2023, with a maximum amplitude reaching 24%, which is far higher than previously reported events with average amplitudes of 2-3% (Li et al., 2016; Tang et al., 2014; Suzuki et al., 2007a). Through ground-based dual-layer and multi-satellite joint observations, we conducted a comprehensive analysis of these events to reveal their role in vertical energy transfer and atmospheric coupling.

## 2. Ground based Airglow Imager and Satellite observation

### 2.1 Airglow Imager

The airglow imager used to observe CGW is installed at the Southern Space Observatory (SSO), the National Institute for Space Research, in São Martinho da Serra (SMS) (29.44°S, 53.82°W), Brazil. Figure 1 shows the location of the airglow imager station at SMS. The imager has a cooled Charge-Coupled Device (CCD) camera with a Mamiya (Focal Length = 24 mm) fish-eye lens of a 180° field of view (FOV) and a resolution of 512 × 512 pixels. The imager is equipped with a filter wheel, and the wheel rotates to observe OH (Wüst et al., 2023) broadband emission (715–930 nm, with a notch at 865.5 nm to suppress the $O_2(0, 1)$ emission) and $O(^1D)$ (630.0 nm, 2.0 nm), respectively. The time resolution of the OH airglow image is 112 s, while that of the OI 630 nm airglow image is 225 s. The exposure times of the OH airglow image and the OI 630 nm airglow image are 20 s and 90 s, respectively. Airglow observations are conducted when the solar depression angle is less than −12° 。

106

**Figure 1.** The location of the airglow imager station at SMS (red triangle). The circle on the map gives the effective observation ranges of OH airglow imager with a 164° field of view. The red asterisks and blue asterisks denote the TIMED/SABER ascending and descending track footprints passing over SMS on 18 September 2023, respectively.

Before effectively extracting the wave parameters, the raw airglow images need to be processed through the following steps: First, a median filter with a kernel size of $17 \times 17$ pixels was employed to eliminate stars from the raw images (Li et al., 2011). We also removed the CCD dark noise, which was estimated from dark-frame images captured with the shutter closed prior to observations. Second, we corrected for the van Rhijn effect and atmospheric extinction using the approach described in Kubota et al. (2001). The observed airglow intensity $I(\theta)$ from the ground is not

uniform across different zenith angles. This non-uniformity is due to the van Rhijn
effect. Additionally, the observed airglow intensity is influenced by atmospheric
extinction, which results from absorption and scattering along the line of sight.

Since airglow observations are subject to the van Rhijn effect, the measured

emission intensity at a specific zenith angle (θ) follows the relation (Kubota et al.,

2001):

$$I(\theta) = I(0) \cdot V(H, \theta),$$


$$V(H, \theta) = \left[1 - \left(\frac{R}{R+H}\right)^2 \sin^2(\theta)\right]^{-\frac{1}{2}}, \tag{1}$$

where $I(0)$ is the emission intensity at zenith. $V(H, \theta)$ is the van Rhijn correction
factor. $R$ is the earth radius and $H$ is the height of OH airglow layer. The relationship
between the observed emission intensity $I(\theta)$ —affected by atmospheric extinction—
and the true emission intensity $I_{true}(\theta)$ at the airglow layer is described by Kubota
et al. (2001).

$$I(\theta) = I_{true}(\theta) \cdot 10^{-0.4 \cdot a \cdot F(\theta)},$$


$$F(\theta) = [\cos\theta + 0.15 \cdot (93.885 - \theta \cdot \frac{180}{\pi})^{-1.253}]^{-1}, \tag{2}$$

where a is the atmospheric extinction coefficient, $F(\theta)$ is an empirical equation.

Consequently, the image correction factor, obtained from the combination of

Eqs. (1) and (2), takes the form:

$$K = V(H, \theta) \cdot 10^{-0.4 \cdot a \cdot F(\theta)}. \tag{3}$$

The parameter a depends on the atmospheric observing conditions. For the observed
CGW events, we treat a as temporally constant. By averaging the images over the
observation period, we derive the zenith-angle-dependent airglow intensity profile.
The optimal value of $a$ is determined by matching this observed profile with
theoretical $K$ profiles across varying $a$. The fitted value of parameter $a$ is
approximately 0.42. Finally, we apply the flat-field correction by dividing the raw
images by the corresponding $K$ factor.
Third, we eliminated atmospheric background counts from the images. For
background emission, Swenson and Mende (1994) used simultaneous Infrared
measurements to demonstrate that the background contributes approximately 30% of
the total OH airglow image signal. Similarly, Suzuki et al. (2007b) confirmed this
ratio (~30%) through concurrent OH intensity observations with a Spectral Airglow
Temperature Imager. In this study, we adopt the same assumption that background
emissions account for ~30% of the total signal.
Then, the original airglow images were spatially calibrated using stars as
reference points. Each pixel location (i, j) in the original image was first mapped to a
position (f, g) in a standardized coordinate system. Subsequently, the point (f, g) was
transformed into geographic coordinates (x, y) using azimuth (az) and elevation (el)
angles.
The conversion between original image coordinates (i, j) and standard
coordinates (f, g) is defined by a linear transformation (Hapgood and Taylor, 1982):
$$\begin{bmatrix} f \\ g \end{bmatrix} = \begin{bmatrix} a_0 & a_1 & a_2 \\ b_0 & b_1 & b_2 \end{bmatrix} \begin{bmatrix} 1 \\ i \\ j \end{bmatrix},$$
(4)

where the coefficients a and b are calculated by applying a least-squares fitting using
the observed location of the stars in the original image and their locations in standard

coordinate (Garcia et al., 1997):

$$\begin{bmatrix} a_0 & b_0 \\ a_1 & b_1 \\ a_2 & b_2 \end{bmatrix} = \begin{bmatrix} \mathbf{1}^T\mathbf{1} & \mathbf{1}^T\mathbf{i} & \mathbf{1}^T\mathbf{j} \\ \mathbf{1}^T\mathbf{i} & \mathbf{i}^T\mathbf{i} & \mathbf{i}^T\mathbf{j} \\ \mathbf{1}^T\mathbf{j} & \mathbf{i}^T\mathbf{j} & \mathbf{j}^T\mathbf{j} \end{bmatrix}^{-1} \begin{bmatrix} \mathbf{1}^T \\ \mathbf{i}^T \\ \mathbf{j}^T \end{bmatrix} \begin{bmatrix} \mathbf{f} & \mathbf{g} \end{bmatrix}, \tag{5}$$

where the column vectors $\mathbf{i}$ and $\mathbf{j}$ contain observed star locations in the original image, while $\mathbf{f}$ and $\mathbf{g}$ hold their computed normalized coordinates. The vector $\mathbf{1}$ is a constant-valued column vector with length matching these vectors.

Through a georeference procedure, the standard coordinate images were projected onto geographic coordinates, assuming peak emission heights of 87 km for the OH layer and 250 km for the OI 630.0 nm layer. The spatial resolution of the imager varies significantly zenith angle. For the OH channel, it is 0.53 km/pixel at the center of the image and degrades to 39.8 km/pixel at the edge of the image. For the 630 channel, the resolution is 1.53 km/pixel at the center of the image and decreases to 40.8 km/pixel at the edge of the image.

**2.2 GOES, Aqua, Suomi NPP, and TIMED Satellite Observations**

**2.2.1 GOES Satellite Observations**

The Geostationary Operational Environmental Satellite-16 (GOES-16) (Schmit et al., 2005), launched in November 2016, is part of the GOES-R Series. The Advanced Baseline Imager (ABI) is the primary instrument on GOES-16, providing high-resolution imagery in 16 spectral bands, including 2 visible channels (0.47 μm and 0.64 μm), 4 near-infrared channels (0.86 μm, 1.37 μm, 1.6 μm, and 2.2 μm), and 10 infrared channels (3.9–13.3 μm), with a temporal resolution of 10 min and a spatial resolution of 0.5–2 km (Schmit et al., 2017). The brightness temperature (BT), derived from 10.3 μm infrared images from channel 13, is used

to study the convection activities during the CGW events.
**2.2.2 Aqua Satellite Observations**
The Atmospheric Infrared Sounder (AIRS) (Aumann et al., 2003; Chahine et al.,
2006) is an infrared spectrometer and sounder onboard the NASA Aqua satellite
(Parkinson et al., 2003). AIRS performs continuous across-track scanning, acquiring
data footprints sequentially. The collected data are then organized into 6-minute
granules. The footprint size of AIRS is approximately 13–14 km in diameter at nadir
view, and the scan swath width is around 1765 km (Hoffmann et al., 2014). AIRS
is capable of detecting air thermal perturbations induced by GWs with vertical
wavelengths longer than 10–15 km and horizontal wavelengths ~50–500 km
(Hoffmann and Alexander, 2010). The radiance measurements at the 4.3 μm $CO_2$
fundamental emission band are particularly sensitive at altitudes around 30–40 km.
In this study, the $CO_2$ radiance emission band with frequencies ranging between
2299.80 $cm^{-1}$ and 2422.85 $cm^{-1}$ (Rothman et al., 2013) is utilized to measure
stratospheric air temperature perturbations.
**2.2.3 Suomi NPP Satellite Observations**
The Visible Infrared Imaging Radiometer Suite (VIIRS) instrument, onboard
the Suomi NPP satellite (Lee et al., 2010; Lewis et al., 2010), is a multispectral
scanner capable of capturing high-resolution images in both visible and infrared
wavelengths. The Day Night Band (DNB) of the VIIRS sensor operates in the
visible/near-infrared (NIR) range, covering wavelengths from 500 to 900 nm
(Miller et al., 2012), which includes three key mesospheric airglow emissions: the

O(1S) line at 557.7 nm, the Na doublet at 589.0/589.6 nm, and the OH Meinel

band (~600–900 nm). The sensor has a high spatial resolution of 0.375 km at nadir

for its imagery bands and 0.75 km for its moderate-resolution bands. The VIIRS

sensor has a wide across-track swath width of 3000 km.

### 2.2.4   TIMED Satellite Observations

Sounding of the Atmosphere using Broadband Emission Radiometry (SABER)

is one of four instruments on NASA's Thermosphere Ionosphere Mesosphere

Energetics Dynamics (TIMED) satellite (Russell et al., 1999), launched on December

7, 2001. TIMED focuses on exploring the energy properties and redistribution in the

MLT region, providing data to define the basic states and thermal balance of this area.

SABER is a 10-channel broadband limb-scanning infrared radiometer (1.27-17 μm).

It measures kinetic temperature through $CO_2$ emissions (15 μm Local

Thermodynamic Equilibrium (LTE) below 90 km; 4.3 μm non-LTE above 90 km)

with ±2-5 K accuracy. Simultaneously observing $O_3$ (9.6 μm), OH (1.6-2.0 μm), and

$O_2$ (1.27 μm) emissions, it quantifies radiative cooling (up to 150 K/day) and chemical

heating (~8 K/day) in the MLT region with 2-4 km vertical resolution.

## 3   Observations

### 3.1 Double-layer All sky Airglow Imager Observations
### 3.1.1 Mesospheric Concentric Gravity Waves from OH All sky imaging observation

Three groups of intense CGWs (wave packets nos. 1–3) were captured by the

OH emission channel of the airglow imager at the Southern Space Observatory (SSO)

in São Martinho da Serra (29.44°S, 53.82°W) in southern Brazil on 17-18 September

2023. These events initially emerged within the imager's field of view at 22:25:02 UT

on 17 September and remained continuously detectable until the cessation of
observational recording at 08:35:15 UT on 18 September, thereby spanning an
extended duration in excess of 10 hours. For more detailed information on the wave
propagation status, please refer to the Supplement (http://doi.org/10.5446/69990, Li,
2025a). Figure 2 shows the time sequence of CGW no. 1 from 22:49:23 UT on 17
September to 03:39:31 UT on 18 September. CGW no. 1 first appeared in the
southwest direction of the station.
The distinct visible concentric wavefronts radiating outward from the center (red
dot in each panel) are indicative of the atmospheric response to disturbances caused
by strong convection in the lower atmosphere. Interestingly, the center of CGW no. 1
continues to move eastward. Between 22:45:38 UT on 17 September and 05:26:13
UT on 18 September, the center moved approximately 436 km eastward, with an
average speed reaching ~65 km/h. This eastward drift of the wave's center could be
indicative of the influence of prevailing wind patterns and the eastward movement of
the convective system itself. The horizontal wavelengths of the GWs at radii of 0–
300 km (denoted by the red line in Fig. 2 at 23:39:55 UT) are measured to be (30–82)
$\pm$ 3 km. The observed period is $9.0 \pm 3.5$ min, and the observed phase speed is 80–
110 ms$^{-1}$. In the northwest direction (denoted by the red line in Fig. 2 at 00:49:11 UT),
we have detected larger-scale waves with a wavelength of about 160 km, a period of
approximately 16 min, and a phase speed of about 167 ms$^{-1}$.

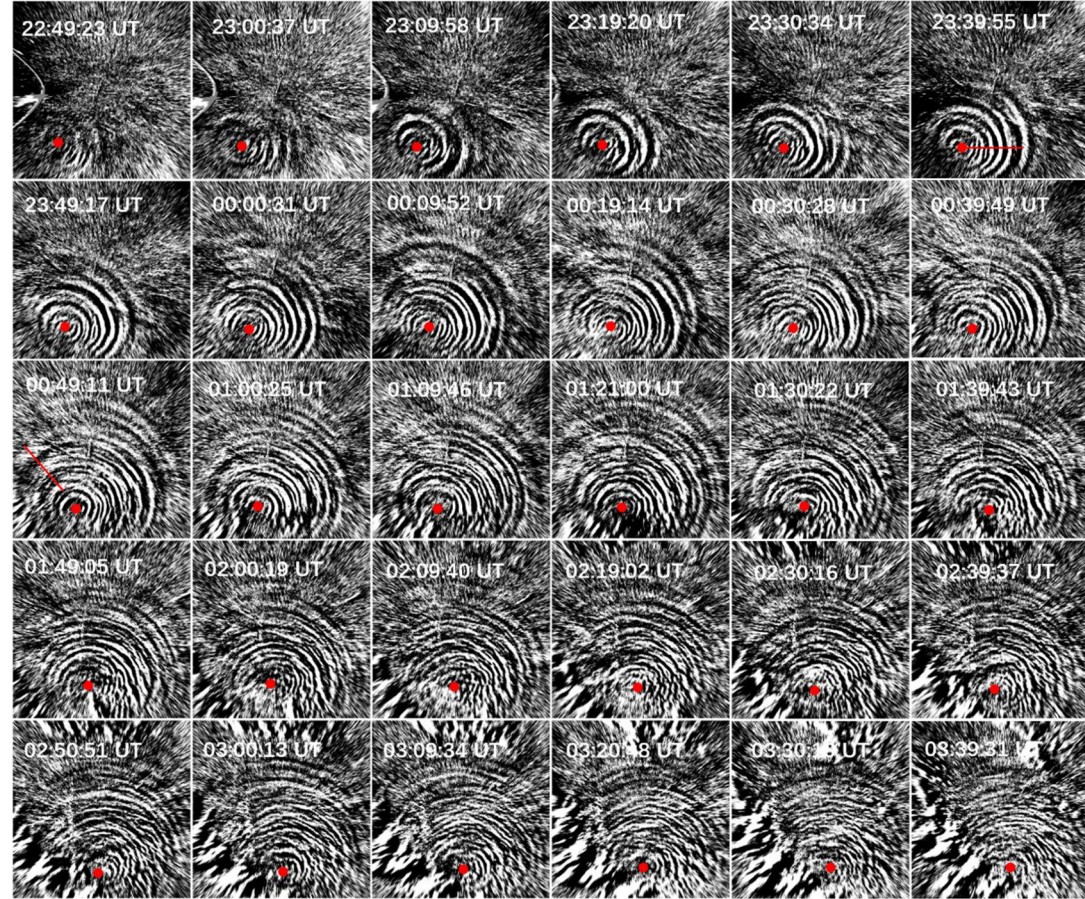

**Figure 2.** All-sky OH images projected onto an area of 1000 km×1000 km showing the CGW no.1 event at half-hour intervals in the SMS station on 17-18 September 2023. The red dots mark the estimated centers of the CGW. The presented images display the corrected OH emission intensity.

From 02:00 UT, clouds began forming in the southwestern and western sectors of the station (see Fig. 2). By 04:00 UT, cloud formation extended to the zenith and northern sectors, persisting until ~05:30 UT. Figure 3 shows the time sequence of CGW no. 2 and CGW no. 3 from 03:58:14 UT on 17 September to 07:59:42 UT on 18 September. Despite cloud cover, CGW no. 2 and CGW no. 3 were observed in cloud gaps over the western sector at approximately 03:45:08 UT and 05:13:06 UT, respectively. For CGW no. 2, horizontal wavelengths range from 22 to 38 km, with a period of $7 \pm 1.5$ min and a phase speed of 60–78 ms$^{-1}$. CGW no. 3 exhibits

wavelengths of 24–36 km, a period of $6.5 \pm 1.0$ min, and a phase speed of 72–81 ms$^{-1}$.

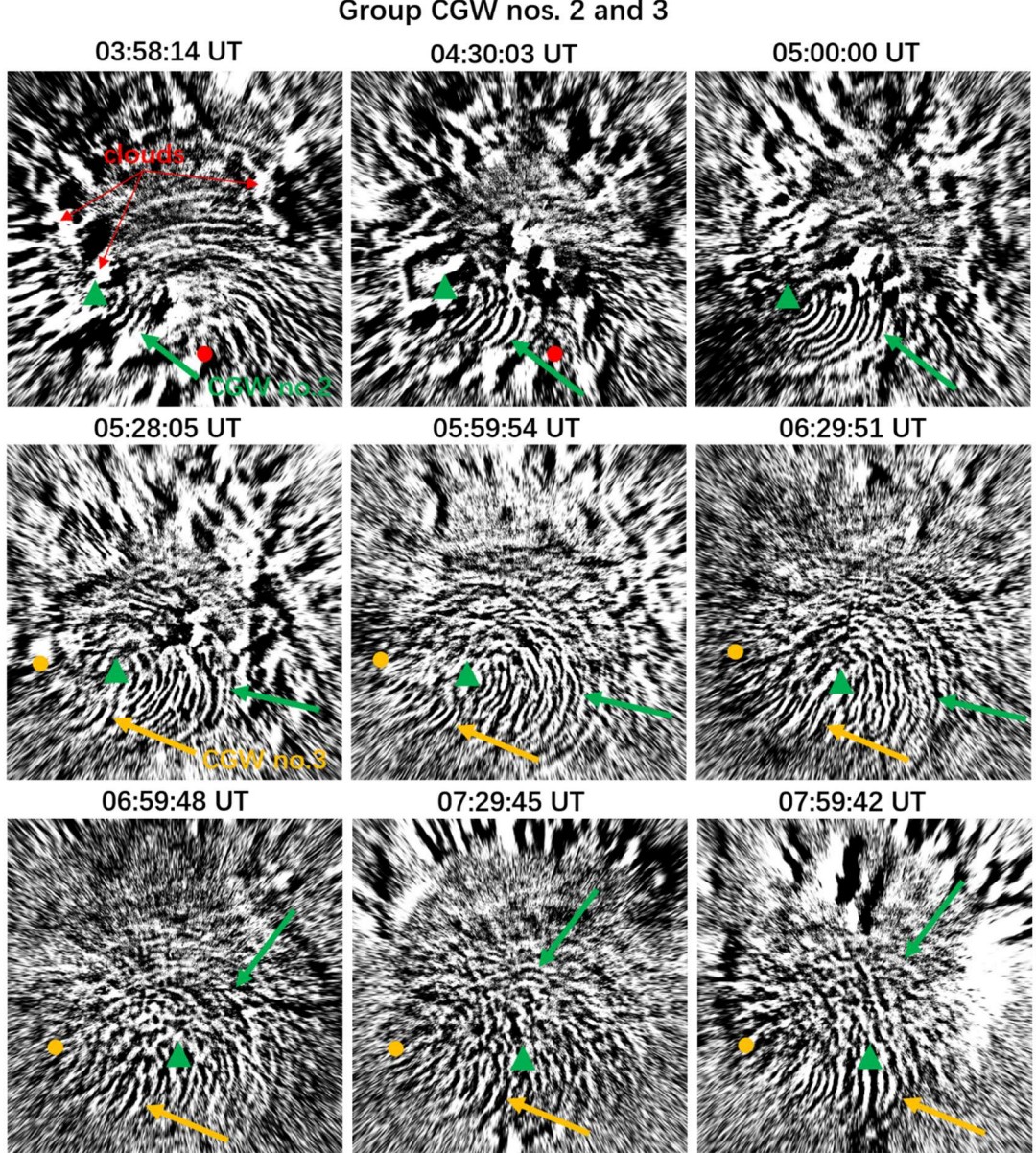

**Figure 3.** All-sky OH images projected onto an area of 1000 km × 1000 km showing the CGW no. 2 and CGW no. 3 events at half-hour intervals in the SMS station on 18 September 2023. The red dot marks the estimated center of the CGW no. 1, while the green and light blue dots indicate the estimated centers of the CGW no. 2 and CGW no. 3, respectively. The presented images display the corrected OH emission intensity.

### 3.1.2 Thermospheric Concentric Gravity Waves from All sky 630.0 nm imaging observation

The 630.0 nm filter used in the imager is a narrowband interference filter with a

central wavelength of 630.0 nm and a full-width at half-maximum (FWHM) spectral
width of 2.0 nm. Three spectral lines from the OH (9–3) band lie within the bandwidth
of the 630.0 nm filter: the P2(3) line at 629.7903 nm, the P1(3) doublet at 630.6869
nm and 630.6981 nm, and the P1(2) line at 628.7434 nm (Hernandez, 1974; Burnside
et al., 1977; Smith et al., 2013). To determine whether the OI 630 nm airglow image
is contaminated by OH airglow emission, we project both the OH airglow image and
the OI 630 nm airglow image onto the height of the OH airglow layer. We can clearly
see that the OI 630 nm airglow image is contaminated by OH emission, with the
CGWs observed in the OH airglow layer being superimposed onto the OI 630 nm
airglow image denoted by the yellow dashed boxes in Fig. 4. Thus, we must exercise
extreme caution when interpreting disturbances in the thermosphere observed at the
630 nm wavelength, particularly in the absence of concurrent OH airglow
measurements to differentiate whether these disturbances are genuinely
thermospheric phenomena or merely artifacts resulting from OH airglow radiation
contamination. Notably, thermospheric CGW nos.1 and 2 (top panel of Fig. 4) were
unambiguously observed. Their spatial mapping onto OH images confirms these
signals originate from the thermosphere (bottom panel of Fig. 4), excluding OH
contamination. Regarding the contamination of 630 nm images by OH emissions and
the actual propagation situations of CGWs in the thermosphere, please refer to the
Supplement (http://doi.org/10.5446/69989, Li, 2025b).

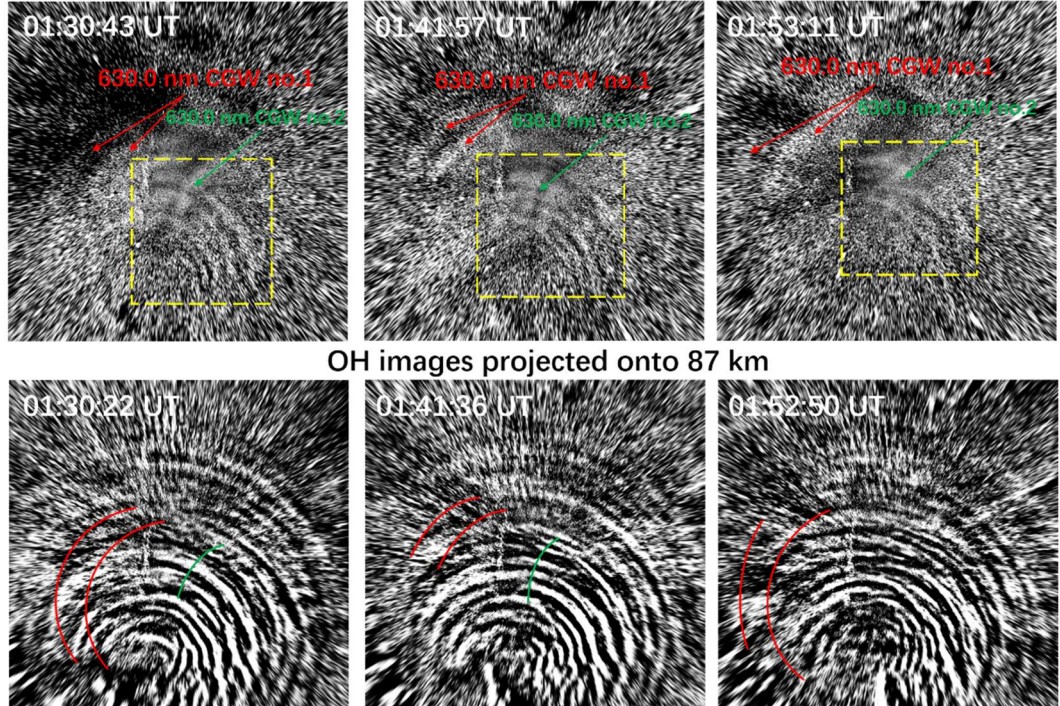

**630.0 nm images projected onto 87 km**

01:30:43 UT   01:41:57 UT   01:53:11 UT

**OH images projected onto 87 km**

01:30:22 UT   01:41:36 UT   01:52:50 UT

**Figure 4.** All-sky 630.0 nm images (top panel) and OH images (bottom panel) were both projected onto an altitude of 87 km with an area of 1000 km × 1000 km. The northeastward-propagating CGW (marked with a yellow dashed box) shows contamination from OH airglow emission. Thermospheric CGWs propagating northwestward confirmed in 630.0 nm images (top panel). The phase fronts of the thermospheric CGW nos. 1 (red lines) and 2 (green lines) are superimposed onto the OH images (bottom panel).

Figure 5 presents a series of OI 630 nm airglow emission images projected onto an altitude of 250 km. The ring-shaped arc (thermospheric CGW no. 1) (indicated by red arrows) propagating towards the northwest was identified, with a wavelength of approximately 165 km and a horizontal observed phase speed of about 183 ms$^{-1}$. There are also observed curved wave structures (thermospheric CGW no. 2) (indicated by green arrows) whose wave fronts are perpendicular to those of the contaminating OH wave fronts. The optical signatures of medium-scale traveling ionospheric disturbances (MSTIDs) in the southern hemisphere, as observed in OI

630.0 nm emission images, typically manifest as alternating dark and bright bands aligned along the northeast-southwest direction, propagating in a northwestward direction (Candido et al., 2008). The MSTIDs generally exhibit full FOV coverage, traversing the entire imaging region during their propagation. However, our observations revealed that the thermospheric disturbances first emerged in the zenith region, exhibiting distinctively arcuate phase fronts, suggesting that they were excited by a quasi-point source in the lower atmosphere. The fitted center of the arc (indicated by a red dot) is located ~320 km to the southwest of the station.

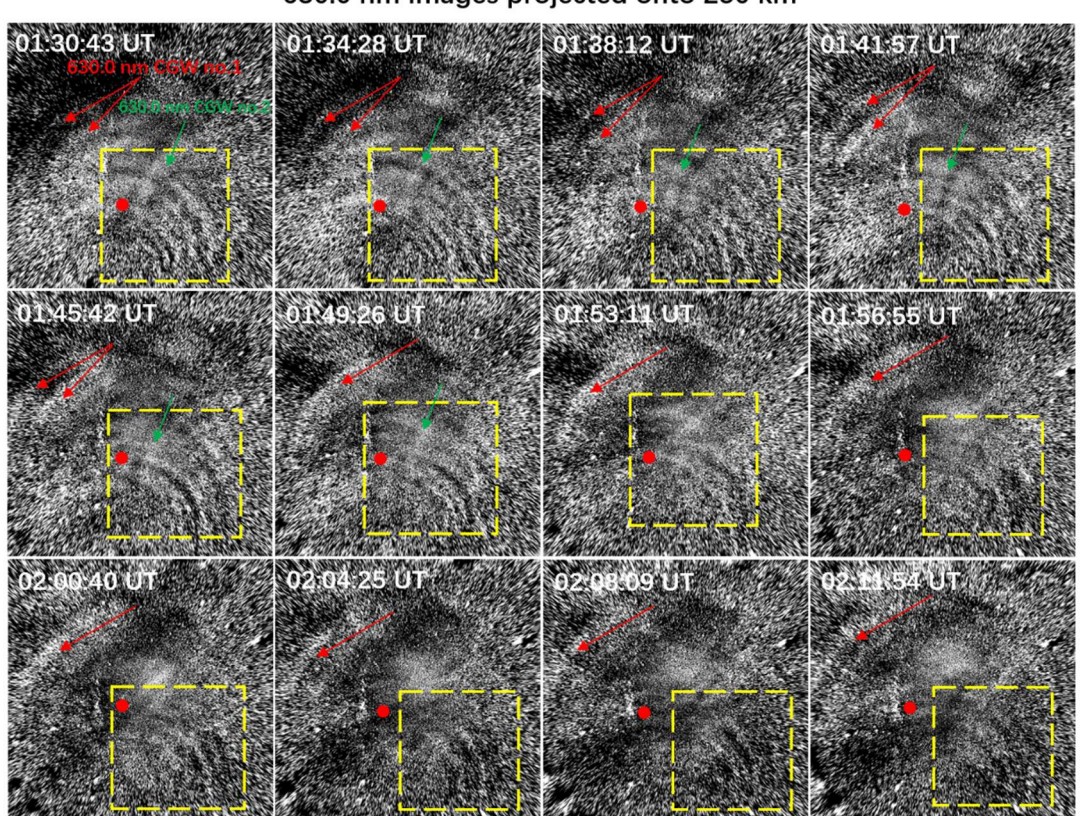

**Figure 5.** All-sky 630.0 nm images projected onto an area of 2000 km × 2000 km showing the thermospheric CGW nos.1 (indicated by red arrows) and 2 (indicated by green arrows) at approximately 4 min intervals in the SMS station on 18 September 2023. The red dots mark the estimated centers of the thermospheric CGW. The northeastward-propagating CGW (marked with a yellow dashed box) exhibits

artifacts influenced by OH airglow emission.
**3.2 AIRS and Suomi NPP**
Figure 6 shows the AIRS 4.3 μm BT perturbation map over southern Brazil
at 05:05:21 UT on 18 September 2023. The AIRS observation reveals large-scale
waves propagating northwestward and westward, with a horizontal wavelength of
approximately 160 km. The limited spatial resolution of AIRS restricts its
detection capability for GWs with short horizontal wavelengths. The relatively
weak brightness temperature fluctuations observed by AIRS may result from the
instrument's limited sensitivity to short vertical wavelengths (Hoffmann et al., 2014).
Consequently, the observed brightness temperature amplitudes are typically much
lower than the actual stratospheric temperature fluctuations, especially for convective
wave events with short vertical wavelengths. Based on the stratospheric CGW's
central position and propagation characteristics, we infer that this wave shares the
same source with mesospheric CGW no. 1 identified in the OH all-sky images.

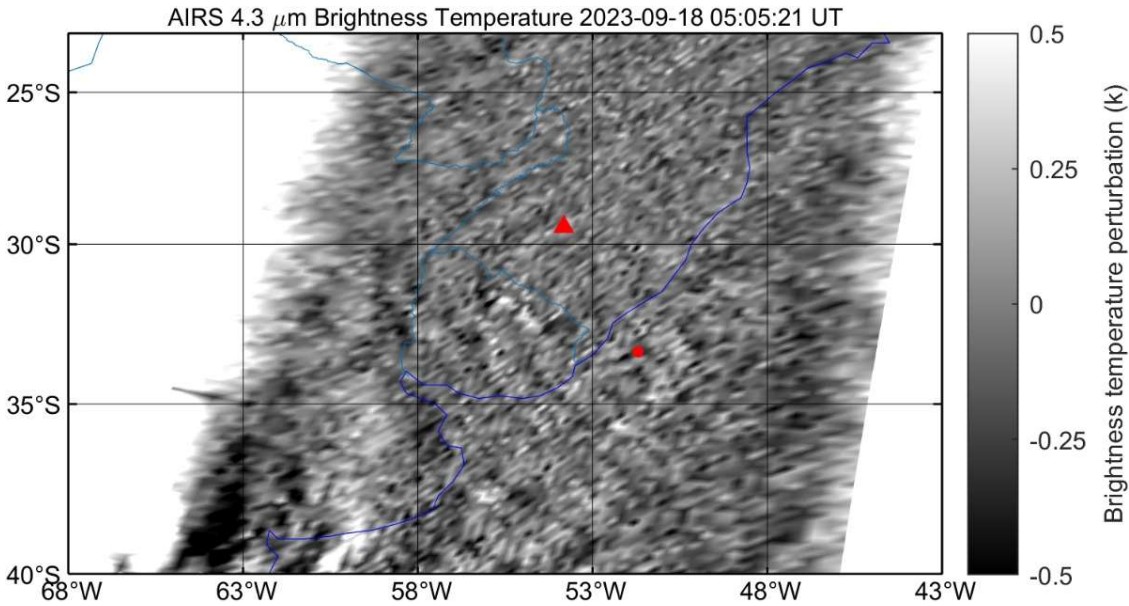

**Figure 6.** Aqua satellite 4.3 μm brightness temperature observations of CGWs at 05:05:21 UT on
18 September 2023. Brightness temperature is derived from 4.3 μm radiance at an altitude range
of 30–40 km. The red triangle and dot mark the SMS station and fitted wave center, respectively.
The Suomi-NPP satellite flew over Southern Brazil region during the
progression of the CGW events. Figure 7 shows CGWs from the S-NPP
VIIRS/DNB band measurements at 03:59:54 UT on 18 September 2023. The
horizontal wavelengths are primarily distributed within the range of (38–52) ± 3
km (indicated by a red dashed box). In the eastern direction of the small-scale
wave region, large-scale waves located at (34°S–39°S, 43°W–46°W) were
detected with a horizontal wavelength of approximately 154 km ± 5 km. Due to
the interference of urban lighting, the CGW structures were not visible over the
land.

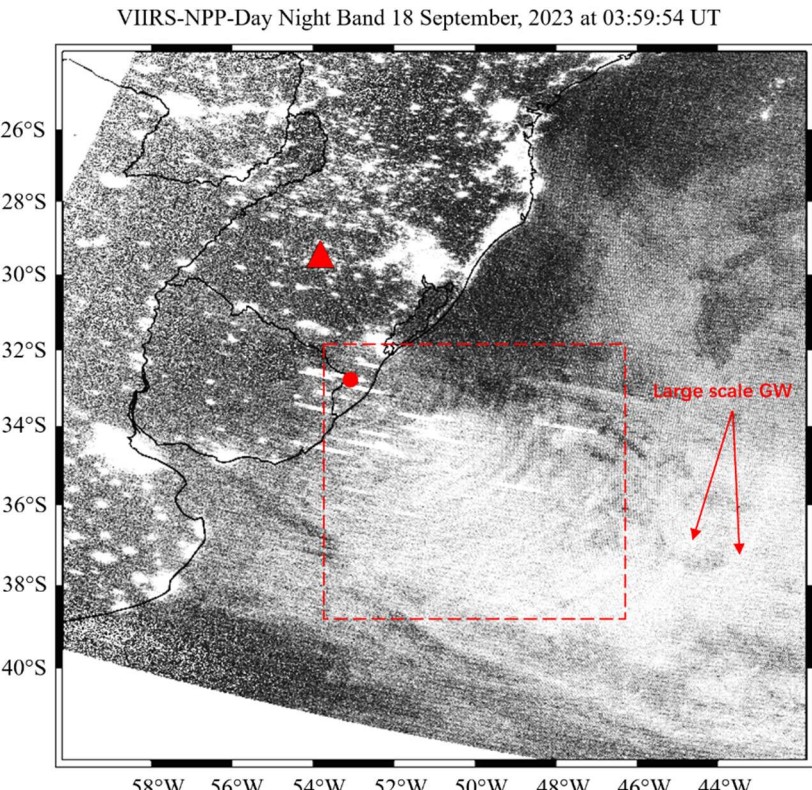

**Figure 7.** Suomi-NPP satellite Day Night Band radiance observations of CGWs at 03:59:54 UT
on 18 September 2023. Red triangle represents the SMS station, and the red dot represents the
position of the fitted center of the CGW.

**3.3 GOES Observations of Convective Plumes**

Figure 8 shows GOES-16 10.3 μm BT over southern Brazil from 21:00 UT to 05:30 UT on 17-18 September 2023. The first convective system initially appeared in the southwest direction of the station (indicated by the red arrow) at around 21:00 UT. This convective system continued to move eastward over time and had traveled approximately 400 kilometers by 05:30 UT. This eastward motion explains the observed ~436 km displacement of CGW no. 1 in the mesopause region. The second and third convective systems appeared at approximately 02:30 UT and 04:30 UT, respectively, and also moved eastward. By 06:30 UT, the three convective systems had merged together. The detailed evolution process of thunderstorm systems is provided in Supplement (http://doi.org/10.5446/69993, Li, 2025c). The spatial proximity of the three CGW centers to the initiation points of the convective systems strongly suggests these systems served as excitation sources for the CGWs detected by the airglow imager.

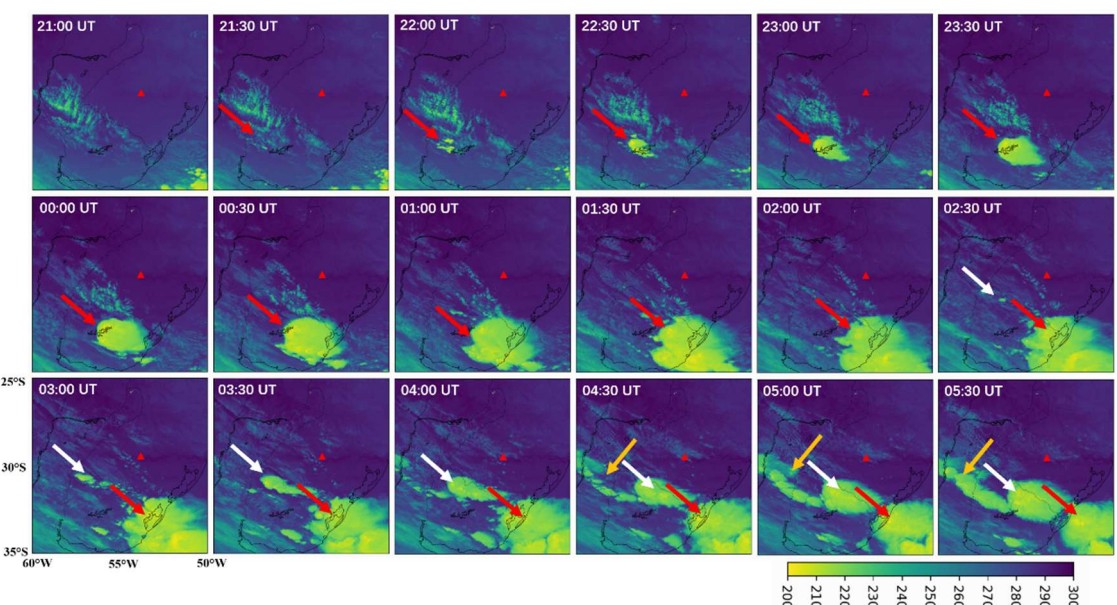

**Figure 8.** GOES-16 10.3 μm brightness temperature from 21:00 UT to 05:30 UT on 17-18

September 2023. The brightness temperature is derived from 10.3 μm infrared radiance data from channel 13. Red triangle represents the SMS station.

## 4   Results and Discussion

### 4.1 The characteristics of mesopause CGWs

We analyzed the background wind field above the station using a composite dataset: the European Centre for Medium-Range Weather Forecasts (ECMWF) ERA5 (Hersbach et al., 2020) for 0-70 km altitude and the Horizontal Wind Model 2014 (HWM14; Drob et al., 2015) for 70-87 km altitude. Figure 9a and b show the zonal wind and meridional wind fields, respectively. Figure 9c presents a critical level filtering diagram, demonstrating how gravity waves from the lower atmosphere are prevented from reaching the mesopause region when their phase velocities fall within the prohibited range. The diagram reveals a maximum blocking amplitude of approximately 44 ms$^{-1}$. The results indicate that weaker background winds (producing smaller blocking amplitudes) enhance the vertical propagation of CGWs from the lower atmosphere to the mesosphere. Apart from the moving convective system mentioned above,which is a primary cause of the eastward displacement of the CGW center observed at the mesopause, the prevailing winds near 10 km and 55 km in Fig. 9a also significantly contribute to the eastward movement of the CGW center.

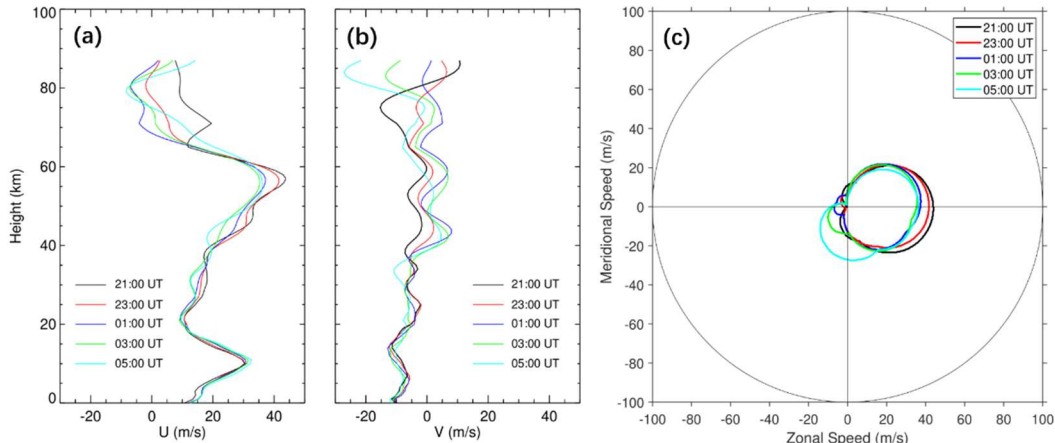

**Figure 9.** (a) The (a) zonal and (b) meridional wind field profiles from ERA5 (0-70 km) and HWM14 model (70-87 km) at 21:00 UT, 23:00 UT, 01:00 UT, 03:00 UT, and 05:00 UT, respectively. (c) Two-dimensional blocking diagrams from 0 to 87 km derived from the wind profiles in (a) and (b) on 17-18 September 2023.

Figure 10 shows sequential cross sections of OH emission intensity perturbations perpendicular to the CGW no. 1 fronts. The wave amplitudes observed in this study exhibit significantly stronger perturbations, with a maximum relative amplitude of 24%. In contrast, previous studies have reported average amplitudes that are approximately 2% (Li et al., 2016; Tang et al., 2014; Suzuki et al., 2007a). Additionally, Smith et al. (2020) reported mean-to-peak wave brightness amplitudes of 10%.

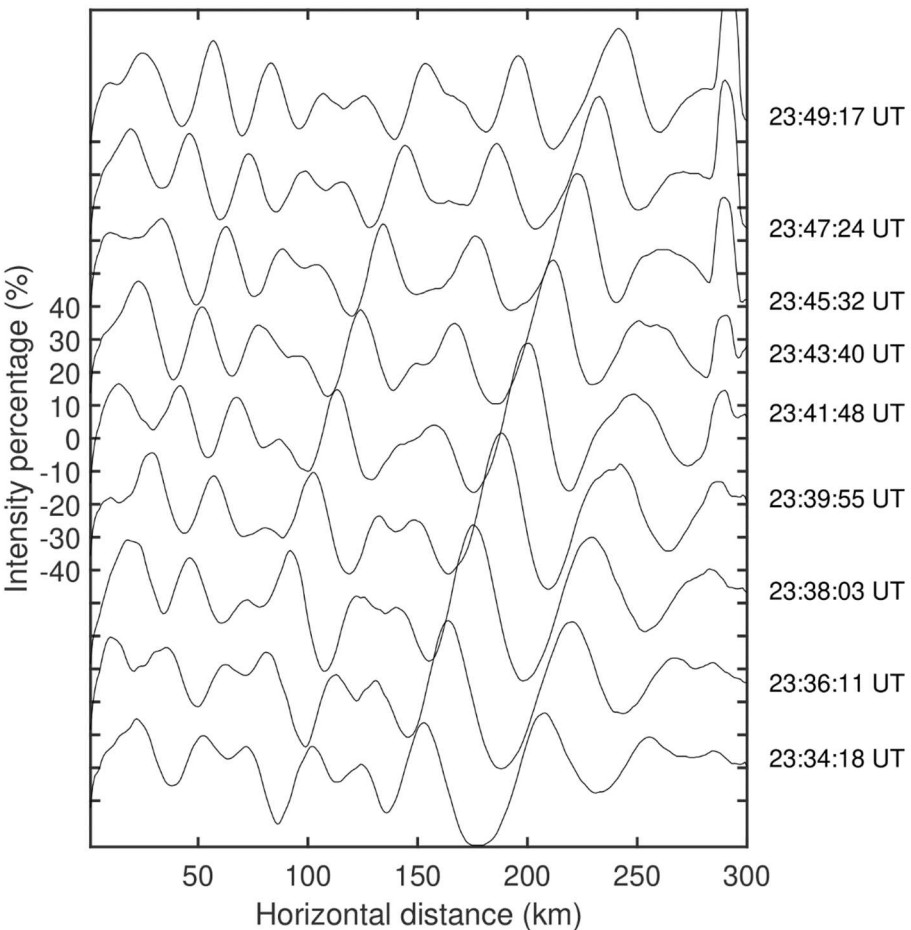

**Figure 10.** OH emission intensity perturbations perpendicular to the CGW no. 1 fronts (denoted by the red line in Fig. 2 at 23:39:55 UT) from 23:34:18 UT to 23:49:17 UT on 17-18 September 2023.

During the generation and propagation of CGWs, two SABER orbits passed over the station and happened to be within the field of view of the airglow imager, as shown in Fig. 11. The first orbit passes over the station at approximately 00:26 UT, followed by a second orbit ~7 hours later at 07:18 UT (Fig. 1). Figure 12 presents seven OH airglow emission and temperature profiles from TIMED/SABER. We observed that the CGWs caused strong disturbances to the airglow layer. We found that the intensity of airglow emission during the first orbit (Fig. 12a) was much stronger than that during the second orbit (Fig. 12c), which

may suggest that the intensity of the fluctuations during the first orbit was much
stronger than that during the second orbit. In addition to this, we also observed a
double-peaked structure in the airglow emission layer. There are weak double-
peak structures during the first overpass at 00:24:10 UT and 00:28:15 UT. In
contrast, the double-peak structure is more prominent during the second overpass
in the 07:18:23 UT profile.

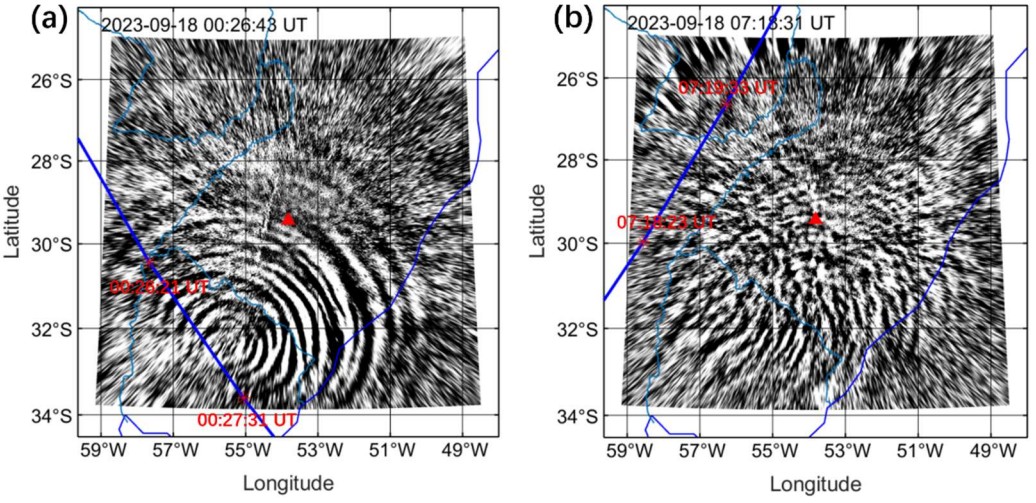


**Figure 11.** Simultaneous observations of mesopause CGWs using OH channel ground-based all-
sky airglow imager and TIMED/SABER satellite measurements. The red triangle marks the
location of the SMS station. The instantaneous field of view of TIMED/SABER is 0.7 mrad by
10 mrad.

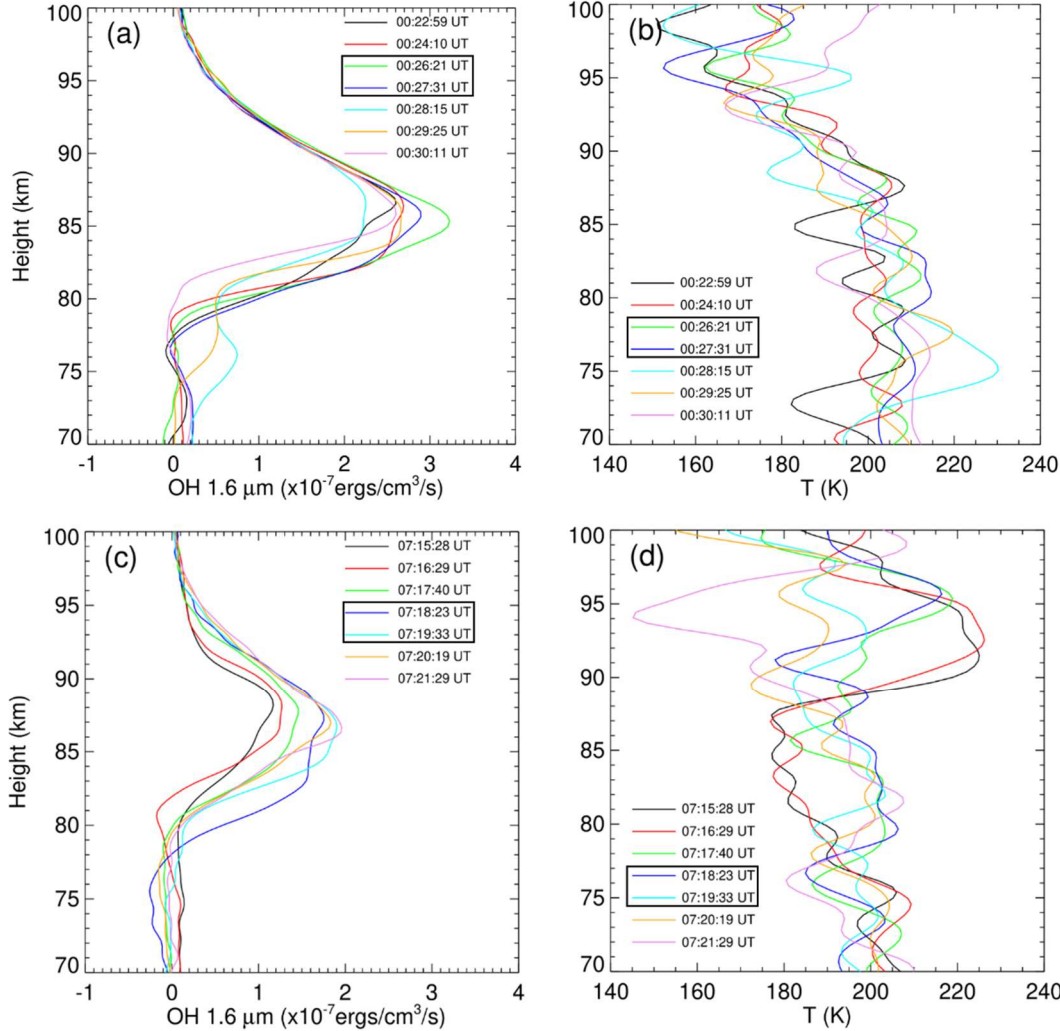

**Figure 12.** TIMED/SABER (a) OH 1.6 μm emission and (b) temperature profiles (ascending

track), and (c) OH 1.6 μm emission and (d) temperature profiles (descending track) on 18

September 2023. Boxed profiles correspond to the satellite's passage through the airglow imager's

effective FOV (see Fig. 11).

We can use airglow imaging observations to estimate gravity wave flux ($F_M$).

The $F_M$ (Swenson and Liu, 1998; Swenson et al., 1999) is expressed as

$$F_M = \frac{1}{2} \frac{g^2}{N^2} \frac{m}{k} \frac{\omega^2}{N^2} \left(\frac{I'}{\overline{I}}\right)^2 \frac{1}{CF^2} \ (m^2 \cdot s^{-2}), \tag{6}$$

where $CF = 3.5 - (3.5 - 0.1)\exp[-0.0055(\lambda_z - 6\mathrm{km})^2]$ is a cancellation factor. $\lambda_z$ is the

vertical wavelength. $I'$ is the perturbed airglow intensity. $\overline{I}$ is the averaged airglow

intensity. g is the gravitational acceleration. $N$ is the Brunt-Väisälä frequency

derived from TIMED/SABER observations. $k = \dfrac{2\pi}{\lambda_h}$ is the horizontal wave number.
$\lambda_h$ is the horizontal wavelength derived from airglow images. $\omega = \dfrac{2\pi c_i}{\lambda_h}$ is the
intrinsic frequency (where $c_i$ is the intrinsic phase speed). $m = \dfrac{2\pi}{\lambda_z}$ is the vertical
wave number derived from the GW dispersion relation (Hines, 1960)
$$m^2 = \frac{N^2}{(c-u)^2} - k^2 - \frac{1}{4H^2}, \tag{7}$$

where $c$ is the observed horizontal phase speed of the wave, $u$ is the wind speed in the
wave direction derived from HWM14, $H$ is the scale height from the SABER
temperature profile.
Figure 13 shows the calculated vertical flux of the horizontal momentum flux of
mesopause CGWs in the altitude of the OH layer from 22:00 to 09:00 UT on 17-18
September 2023. We found that CGW no. 1 produced substantially stronger
momentum flux (peak value >450 m²s⁻²) compared to CGW no. 2 and CGW no. 3,
which showed similar but weaker magnitudes. These values markedly exceed
previous measurements (typically 1-17 m²s⁻² in Li et al. 2016 and Tang et al. 2014)
and even surpass the intense event (decaying from 300 to 150 m²s⁻²) reported by Smith
et al. (2020). Ern et al. (2018) studied the climatology momentum flux determined
from SABER satellite limb sounding data. They find that the GW absolute
momentum flux is approximately 1–4 m²s⁻² in the mesopause region. The results
reveal that the fast-moving thunderstorm systems generated exceptionally powerful
wave activity, transporting substantial momentum and energy into the MLT region.
This demonstrates  remarkable wave coupling between the lower and upper
atmosphere.

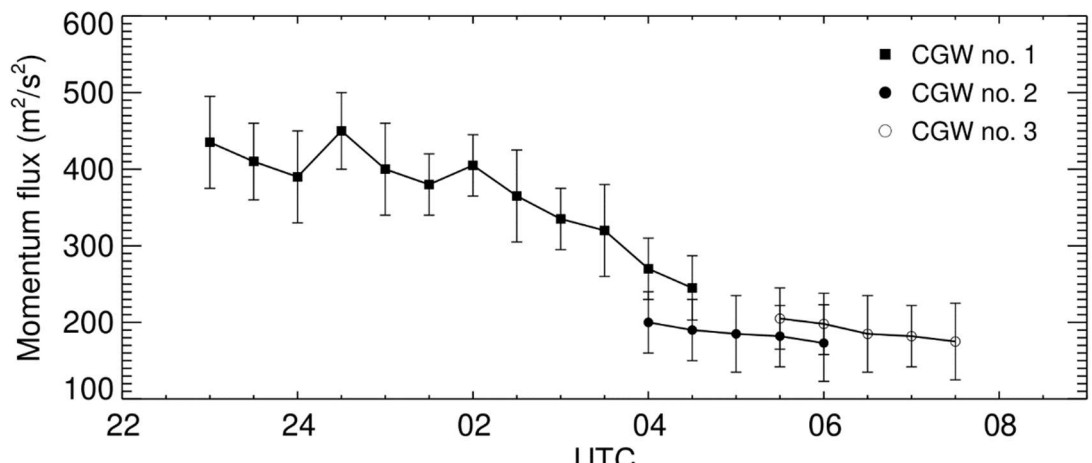


**Figure 13.** Temporal evolution of vertical flux of horizontal momentum from 22:00 to 09:00
UT on 17-18 September 2023.
We use the following vertical group velocity equation to estimate the time
required for the CGWs generated by the convective systems to propagate to the MLT
region.
$$C_{gz} = \frac{\Delta z}{\Delta t} = -\frac{Nkm}{(k^2 + m^2)^{3/2}},$$   (8)
where $\Delta z$ and $\Delta t$ are the vertical distance and propagation time of the CGWs from
the troposphere to the airglow layer, respectively. The horizontal wavenumber $k$ is
derived from airglow images. The Brunt-Väisälä frequency $N$ and vertical
wavenumber $m$ were calculated as the mean value over the atmospheric layer
spanning from the tropopause to the mesopause. Notably, the background wind and
temperature may exhibit significant altitudinal variations, resulting in substantial
variations in the CGW vertical group velocity.
The background temperature for calculating the vertical group velocity of CGW
no. 1, no. 2, and no. 3 was derived from TIMED/SABER profiles within effective
FOV of the OH imager during the first orbit (Fig. 12b), the average of the first and
second orbits (Fig. 12d), and the second orbit, respectively, while wind field data
combined ERA5 (0–70 km) and HWM14 (70–87 km). The vertical group velocies
of CGW no. 1, CGW no. 2, and CGW no. 3 are estimated to be 27–42 ms$^{-1}$, 21–32
ms$^{-1}$, and 24–31 ms$^{-1}$, respectively. This implies that the time taken for CGW no. 1,
CGW no. 2, and CGW no. 3 to reach the OH airglow layer (87 km) is approximately
28-44 min, 37-57 min, and 38-50 min, assuming the excitation height of CGWs is 15
km. Yue et al. (2013) conducted multilayer observations of convective gravity waves
over the western Great Plains of North America and estimated that the time from the
convective source to the airglow layer was ~45 min.
**4.2 The characteristics of thermospheric CGWs**
We further investigated the propagation characteristics of thermospheric CGW
no. 1. The vertical group velocity of the thermospheric gravity waves can be estimated
using the following approximate relationship: $C_{gz} \sim -\dfrac{N}{k}\cos^2\alpha\sin\alpha$ . $\alpha$ is zenith
angle between the vertical altitude and propagation direction of the CGWs phase
fronts. The zenith angle $\alpha$ is approximately 61° from Fig. 14a. The buoyancy
frequency N is estimated to be $2\pi/10.35$ min at the thermosphere height of 250 km,
which is derived from the empirical neutral atmosphere model (NRLMSISE-00)
(Picone et al., 2002). The horizontal wavenumber k=$2\pi/165$ km. The estimated
vertical group velocity is about $54 \pm 6$ ms$^{-1}$. Based on the vertical group velocity, we
find that the time taken for the gravity waves to propagate from the OH layer and the
tropause region to the thermosphere is approximately $50 \pm 5$ min and $73 \pm 8$ min,
respectively. As discussed above, the OH images and OI images were captured nearly
simultaneously to illustrate the contamination effect in Fig. 4. Some of the wave
pattern mismatches in Fig. 4 are due to the propagation time required for CGWs to
travel from the OH altitude to the OI altitude. Given the thermospheric arrival time
of 01:41:57 UT (Fig. 14a), the CGWs were likely excited near the tropopause ($\sim$15
km altitude) at approximately 00:28:57 UT (Fig. 14c), passed through the OH layer
($\sim$87 km altitude) between approximately 00:46:57 UT and 00:56:57 UT. Notably,
GWs with comparable scales were observed in the OH layer at around 00:54:48 UT
(Fig. 14b), which suggests that they might be the same wave.

As mentioned above, the observed thermospheric CGWs exhibits an asymmetric

structure, appearing as arc-shaped waves only in the western and northwestern
directions. This asymmetry can be attributed to the Doppler effect of the background
wind field, which influences gravity wave detection through wave cancellation. GWs
propagating against background wind are Doppler shifted to a larger vertical
wavelength, and increased chance of observation (Li et al., 2016). These GWs suffer
little cancelation and can be easily detected by airglow imager GWs observations.
GWs propagating along background wind are Doppler shifted to a smaller vertical
wavelength, causing the wave amplitude to become invisible. As illustrated in Fig.
14d, the eastward zonal wind at 250 km altitude reaches $\sim$90 ms$^{-1}$. This strong
eastward wind likely suppresses the visibility of eastward-propagating thermospheric
CGWs in airglow imaging. We use Eq. 5 to estimate that the vertical wavelength of
thermospheric CGWs propagating in the northwest direction is approximately 236
km, while that of thermospheric CGWs propagating eastward is approximately 62
km. The Doppler shift reduces their vertical wavelengths, causing them to fall below
the detection threshold of the vertically integrated airglow observations, which is
approximately 100 km from 200 km to 300 km during nighttime (Chiang et al., 2018).

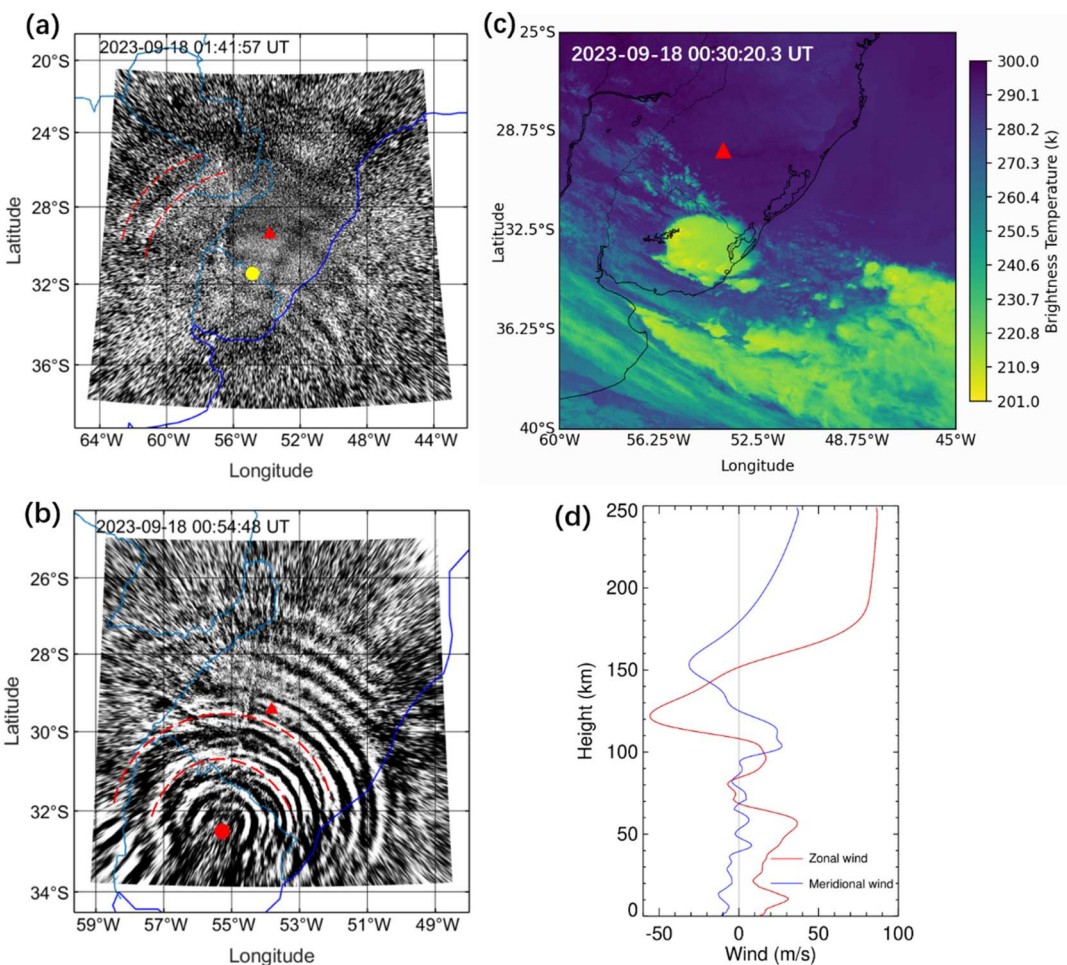


**Figure 14.** (a) All-sky 630.0 nm imaging observation of thermospheric CGW (red dashed lines)
at 01:41:57 UT on 18 September 2023. The yellow dot marks the estimated center of the
thermospheric CGW. (b) All-sky OH imaging observation of mesospheric CGW at 00:54:48 UT
on 18 September 2023. The red dashed lines mark out the mesospheric CGW with the same scale
as the thermospheric CGW. The red dot marks the estimated center of the mesospheric CGW. (c)
GOES-16 10.3 μm brightness temperature at 00:20:20 UT on 17-18 September 2023. The red
triangle marks the location of the SMS station. (d) Wind profiles from ERA-5 (0-70 km) and

HWM14 (70-250 km) averaged between 01:00 UT and 02:00 UT on 18 September 2023.

## 5 Conclusions

In this study, we investigated intense CGWs using coordinated dual-channel airglow observations (630.0 nm and OH bands) from the Southern Space Observatory (SSO) in São Martinho da Serra, Brazil, complemented by multi-satellite measurements during 17-18 September 2023. The key findings are summarized as follows:

These unprecedented CGWs exhibited remarkable persistence (>10 hours), extreme amplitude perturbations (>24%), and substantial wave-center movement (>400 km). These wave events were unambiguously linked to fast-moving convective systems observed by GOES-16. The weaker background wind field during the spring season transition was identified as a crucial factor that allowed CGWs to propagate from the lower atmosphere to the MLT region.

The OI 630 nm airglow observations were substantially contaminated by overlapping OH Meinel band emissions (715-930 nm). This contamination leads to spurious apparent vertical coupling, as mesospheric gravity waves (CGWs) are artificially projected onto the thermospheric OI 630 nm emission layer. This cross-layer aliasing effect necessitates rigorous validation protocols when interpreting putative thermospheric disturbances at 630 nm, particularly requiring spatio-temporally collocated OH airglow measurements (e.g., OH (9–3) bands) to discriminate genuine dynamical processes from lower atmospheric contamination artifacts.

The asymmetric propagation of CGWs in the thermosphere was attributed to
variations in vertical wavelength induced by the Doppler effect of background winds.
Specifically, the eastward zonal wind at 250 km altitude, reaching approximately 90
$ms^{-1}$, reduced the vertical wavelength of eastward-propagating CGWs, making them
undetectable in airglow imaging observations due to vertical integration effects.
This study reveals intense CGWs originating from deep convective systems that
play a dominant role in transferring wave energy and momentum from the
troposphere to the MLT region. These waves exhibited exceptional characteristics
including prolonged persistence, extreme amplitude perturbations, and significant
horizontal **movement**, demonstrating their substantial impact on atmospheric
dynamics and space weather by (1) seeding traveling ionospheric disturbances (TIDs)
that disrupt communications/GPS, (2) triggering plasma instabilities, and (3) altering
thermospheric density, affecting satellite drag.
Our coordinated multi-instrument approach, combining dual-channel airglow
observations with satellite measurements, provides crucial insights into wave
propagation while addressing the challenges of cross-layer contamination in OI 630
nm emissions. These findings significantly advance our understanding of gravity
wave dynamics in the upper atmosphere and establish an improved observational
framework for studying atmospheric coupling processes.

**Data availability.** The airglow data are available from the web page of the Estudo e
Monitoramento    Brasileiro    do    Clima    Espacial    (EMBRACE/INPE)    at

http://www2.inpe.br/climaespacial/portal/en (EMBRACE, 2024). TIMED/SABER

data are accessible from http://saber.gats-inc.com/data.php (Mlynczak et al., 2023).

The ERA5 reanalysis data are available for download from the Copernicus Climate

Change Service Climate Data Store at https://doi.org/10.24381/cds.bd0915c6

(Hersbach et al., 2023). The GOES-16 ABI L1b radiances data are accessible from

https://www.ncdc.noaa.gov/airs-web/search (Schmit et al., 2017).  AIRS radiance

data are accessible from https://disc.gsfc.nasa.gov/datasets/AIRIBRAD_005/

summary (AIRS project, 2007). VIIRS DNB data are distributed by the NOAA

Comprehensive         Large         Array-data         Stewardship         System

(CLASS)(https://www.aev.class.noaa.gov/saa/products/welcome;jsessionid=C3562F

228661BE845B176C9AE2714AE6) (Miller et al., 2012).

**Video supplement.** Extreme mesospheric concentric gravity waves from OH

airglow observations over Southern Brazil is available for view

(http://doi.org/10.5446/69990, Li, 2025a). Thermospheric concentric gravity

waves from OI 630 nm airglow observations over Southern Brazil is available for

view (http://doi.org/10.5446/69989, Li, 2025b). Fast-moving severe thunderstorms

over Southern Brazil from GOES-16 observations is available for view

(http://doi.org/10.5446/69993, Li, 2025c).

**Author contributions.** QL conceived the idea of the article and wrote the manuscript.

JX carried out the analysis of the AIRS and NPP data. XL contributed to the analysis

of the SABER data. YZ contributed to the processing of ECMWF data. WY, XL, HL,
and ZL contributed to the data interpretation and manuscript preparation. CMW and
JVB revised the manuscript. All authors discussed the results and commented on the
paper.

**Competing interests.** The contact author has declared that none of the authors has
any competing interests.

**Acknowledgements.** We thank the National Natural Science Foundation of China
(grant nos. 42374205). The authors thank the Estudo e Monitoramento Brasileiro do
Clima Espacial (EMBRACE/INPE) for the provision of the all-sky data. We
acknowledge the use of data from the Chinese Meridian Project. We appreciate the
TIMED/SABER team for providing the temperature and emission intensity data. We
also thank the European Centre for Medium-Range Weather Forecasts (ECMWF) for
the provision of the ERA5 data and Geostationary Operational Environmental
Satellite (GOES) team for the ABI L1b radiances data. We also thank the NASA
Goddard Earth Sciences Data Information and Services Center (GES DISC) for
providing AIRS data and NOAA Comprehensive Large Array-data Stewardship
System (CLASS) for providing Day Night Band data.

**Financial support.** This research has been supported by the National Natural Science
Foundation of China (grant nos. 42374205) and the Specialized Research Fund of
National Space Science Center, Chinese Academy of Sciences (grant no. E4PD3010).
This work has been supported by the B-type Strategic Priority Program of CAS (grant
no. XDB0780000). The project has also been supported by the Specialized Research
Fund for State Key Laboratories.

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
