# Peer review of "Extreme Concentric Gravity Waves Observed in the Mesosphere"

_EGUsphere, 2025_

## Referee Comment (RC3)

A review on the paper " Extreme Concentric Gravity Waves Observed in the Mesosphere and Thermosphere Regions over Southern Brazil Associated with Fast-Moving Severe Thunderstorms"
by Qinzeng Li, Jiyao Xu, Yajun Zhu, Cristiano M. Wrasse, José V. Bageston, Wei Yuan, Xiao Liu, Weijun Liu, Ying Wen, Hui Li, and Zhengkuan Liu

**General comments**
The paper describes a case study of concentric gravity waves (CGWs) observed with the airglow imager in Brazil on 17-18 September 2023. Also, these CGWs were simultaneously captured by three satellites. Three groups of intense CGWs lasted over 10 hours. The CGWs caused profound airglow emission perturbations exceeding 24%. These CGW events were caused by fast-moving deep convections observed by the GOES-16 satellite. The authors have found that these CGW events represent the most intense vertical transport cases ever recorded, demonstrating remarkable wave coupling between the lower and upper atmosphere.

I have found the paper to be interesting to the atmospheric community. At the same time, I have found a number of issues that should be explained in more detail. That is why I recommend accepting the paper after major revision.

**Specific comments**

Line 71: "…dual-layer airglow observations…"
It is not clear what dual-layer the authors talk about? It should be clarified here.

Line 76: "… across these two atmospheric layers was rare."
Across which two layers?

Lines 86-87: "…São Martinho da Serra…"
Please add Brazil here.

Lines 93-95: "The time resolution of the OH airglow image is 112 seconds, while that of the OI 630 nm airglow image is 225 seconds."
Is it the time resolution or exposure time? What is the exposure time?

Line 98: "…the effective observation ranges of OH airglow imager with a 164°field of view"
Before it was said that "fish-eye lens of a 180° field of view"
What is true?

Lines 101-102: "Before effectively extracting the wave parameters, the raw airglow images need to be processed through the following steps:…"
It is not a complete information on the processing of raw images. Among others, the following steps should be described:
How the atmospheric background was subtracted?
How the dark noise of the sensor was subtracted?
How the flat field correction (non-uniformity of the sensor at different wavelengths) was taken into account?
Was the imager absolutely calibrated in a lab?
Does the imager register airglow intensities in relative or absolute units (Rayleigh) ?
At which solar depression angles does the imager operate?

Line 117: "Third, the processed images were projected onto geographic coordinates,…"
This information is not enough. It should be described in more detail how the optical model of the imager optical system was determined and calculated?
What were the reference points in order to calculate free parameters of the optical model? Stars or lab reference points?
What are the errors of projected pixels in the image center and at the edge of FoV ?
What is the spatial resolution of the imager in the imager center and at the edge of FoV ?

Lines 122-129.
It should be given a reference on the GOES-16 satellite and addressed visible and infrared parameters.

Lines 139-141: "In this study, the $CO_2$ radiance emission band with frequencies ranging between 2299.80 $cm^{-1}$ and 2422.85 $cm^{-1}$ is utilized to measure stratospheric air temperature perturbations."
It should be given a reference on the Aqua satellite and CO2 emissions used in this paper.

Lines 143-144: "The Visible Infrared Imaging Radiometer Suite (VIIRS) instrument, onboard the Suomi NPP satellite…"
It should be given a reference on the Suomi NPP satellite.

Lines 166-167: "CGW no. 1 first appeared in the southeast direction of the station."
Is it in the southeast or in the southwest direction of the station?

Lines 172-175: "…the center moved approximately 436 km westward, with an average speed reaching ~65 km/h. This eastward drift of the wave's center could be indicative of the influence of prevailing wind patterns and the westward movement of the convective system itself."
I hardly understand was it the eastward or westward drift? Or sometimes westward and sometimes eastward? This should be clarified.

Lines 173-174: "This eastward drift of the wave's center could be indicative of the influence of prevailing wind patterns…"
This is very interesting but it is not entirely clear. At what altitude is the prevailing wind considered ? In the tropopause or in the mesopause?

Lines 176-177: "…are measured to be (30–82)±3 km."
Having such a large range of wavelengths what is the physical sense of indicating the error of 3 km ? What does this error tell us? Is it the instrumental error or geophysical wave variability or both?
This is again connected to my above-mentioned questions on What are the errors of projected pixels in the image center and at the edge of FoV ?

In Figs. 2 and 3, what physical quantity can we see on these images? Is it some raw OH emission intensity? Or is it a corrected emission intensity? Or is it OH emission intensity in absolute values? It should be clarified in the figure captions.

In Fig. 3, it is difficult to see the green and light blue dots and arrows for color-blind readers. I recommend changing the green dots to, for example, green triangles.

Caption to Fig.6, please indicate the approximate altitude at which this temperature map is observed.

Line 267: "…horizontal wavelengths are primarily distributed within the range 267 of (38–52) ± 3 km"
Having such a large range of wavelengths what is the physical sense of indicating the error of 3 km ?

Lines 301-307. The sentence is repeated twice, please remove the repeated part.

Lines 306-307: "… when their phase velocities fall within the prohibited range."
What is the prohibited range? How much is it?

Lines 322-327: "We also conducted a statistical analysis of CGWs observed by a meridional airglow observation network across mainland China from September 2023 to August 2024, with data from selected stations including Daicai (25.34°N, 110.34°E), Wendeng (37.18°N, 121.79°E), Mohe (53.48N, 122.34°E), and Naqu (31.73°N, 92.47°E). The results indicate that the average CGW amplitudes ranged between 1.7% and 2.6%."
It seems to me that this is a completely different study, with completely different regions than the area of Brazil under discussion. A reference to this study is needed here. Otherwise it should be removed.

Lines 332-334: "During the generation and propagation of CGWs, two saber orbits passed over the station and happened to be within the field of view of the airglow imager, as shown in Fig. 11."
How is the field of view of TIMED/SABER oriented in Fig.11 ? Please add this information.

Lines 341-342: "In addition to this, we also observed a double-peaked structure in the airglow emission layer."
Which airglow SABER profiles do demonstrate a double-peaked structure? This should be paid attention to.

Lines 342-346: "From the temperature profiles (Fig. 12b and d), we have detected a rich spectrum of vertically propagating waves with vertical wavelengths between 5 km and 20 km, which consists with concurrent airglow and satellite observations of upward-propagating CGWs."
This sentence sounds very strange to me due to the following reasons:
1. There is no information at all about vertical spectrums of gravity waves derived from airglow and satellite observations. All presented data were about horizontal gravity wave patterns. Of course, using the dispersion relation for gravity waves one can derive a vertical wavelength from a horizontal wavelength, but it was not done in the manuscript so far.
2. Each presented temperature profile shows significant vertical variations, i.e., inside the FoV of the imager and outside it, far away from the imager. How can we be 100% sure that these temperature variations are due to CGWs and not other gravity waves?
This sentence should be redeveloped or removed from the manuscript.

Equation 4. What is $\omega$ here and how was it calculated? What is g here?

Lines 365-366: "…u is the wind speed in the wave direction derived from meteor radar,…"
I could not find any information on a meteor radar used in this study. This information should be provided. Is a meteor radar located in the proximity to the imager? What is the accuracy of estimation of the horizontal wind speed from meteor radar data in the wave direction discussed here?

Equation 6. Where is α in this equation?
The authors do not provide information on how they estimated k, m, N parameters in relation to the vertical direction. I assume these parameters were calculated as mean values over the height range from the tropopause to the mesopause. But m and N may significantly vary with altitude, resulting in variations in the GW vertical group velocity (see for example, Fig. 4 in Dalin et al., 2016). This may provide significant deviation of the estimated propagation times. The author should provide a comment on Equation 6.

Lines 388-390: "The vertical group velocities of CGW no. 1, CGW no. 2, and CGW no. 3 are estimated to be 31–37 ms$-1$, 24–30 ms$-1$, and 26–29 ms$-1$, respectively."
What is the source of these estimated ranges of the vertical group velocities? The uncertainty of which parameters affects the uncertainty in the estimation of the group velocities to a greater extent ?

Figure 14. What are the red dashed lines in Fig.14 a and b ? And I assume that the red triangle marks the location of the SMS station. Right?

Lines 463-464: "…demonstrating their substantial impact on atmospheric dynamics and space weather."
Please provide more information on how these waves substantially impact on space weather.

---

## Author Comment (AC1)

The paper "Extreme Concentric Gravity Waves Observed in the Mesosphere and Thermosphere Regions over Southern Brazil Associated with Fast-Moving Severe Thunderstorms" by Li et al. is a thorough and convincing study of gravity wave events observed by airglow imaging over Brazil. It is demonstrated that the gravity waves were likely excited by thunderstorms in the region. Fortunate propagation conditions allowed to observe full ring structures of the convective gravity waves in OH airglow images.

Overall, this study is very interesting and of relevance for the readership of ACP. The paper is well written, and the figures are of good quality. The paper is therefore recommended for publication in ACP after minor revisions.

For specific and technical comments see below.

Thank you very much for your review of our paper "Extreme Concentric Gravity Waves Observed in the Mesosphere and Thermosphere Regions over Southern Brazil Associated with Fast-Moving Severe Thunderstorms". We are honored to receive such a positive evaluation from you, and we are particularly pleased that you consider this study to be comprehensive and convincing, and of significance to the readership of ACP. We also appreciate your comments on the good quality of the paper's writing and figures. We will carefully review every detail of the paper and make improvements to ensure the quality of the paper reaches a higher standard.

SPECIFIC COMMENTS:

(1) l.41: You should add some more general references for convective gravity waves. For example, Fovell et al. (1992), or Piani et al. (2000):

Fovell, R., Durran, D., and Holton, J. R.: Numerical simulations of convectively generated stratospheric gravity waves, J. Atmos. Sci., 49, 1427-1442, 1992.

Piani, C., Durran, D., Alexander, M. J., and Holton, J. R.: A Numerical Study of Three-Dimensional Gravity Waves Triggered by Deep Tropical Convection and Their Role in the Dynamics of the QBO, J. Atmos. Sci., 57, 3689-3702, https://doi.org/10.1175/1520-0469(2000)057%3C3689:ansotd%3E2.0.co;2, 2000.

Reply: Thank you for your suggestion. The recommended references have been incorporated into the text.

(2) l.42: For the jet/front source mechanisms please add the reference Plougonven and Zhang (2014):

Plougonven, R., and Zhang, F.: Internal gravity waves from atmospheric jets and fronts, Rev. Geophys., 52, 33-76, doi:10.1002/2012RG000419, 2014.

Reply: Thank you for your suggestion. The recommended reference has been incorporated into the text.

(3) l.58: You should mention that another method for determining the source location is backward ray tracing of gravity waves, which can also be performed for circular gravity wave patterns. An example is Ern et al. (2022):

Ern, M., Hoffmann, L., Rhode, S., and Preusse, P.: The mesoscale gravity wave response to the 2022 Tonga volcanic eruption: AIRS and MLS satellite observations and source backtracing, Geophysical Research Letters, 49, e2022GL098626, https://doi.org/10.1029/2022GL098626, 2022.

Reply: Thank you for your suggestion. The following description is incorporated into the main text.

"The backward ray tracing method, employed for source location determination, can also be applied to circular gravity wave patterns (Ern et al. 2022)."

(4) l.121: Please provide a reference for the ABI-GOES instrument. For example:

Schmit, T. J., Gunshor, M. M., Menzel, W. P., Gurka, J. J., Li, J., and Bachmeier, A. S.: Introducing the next-generation advanced baseline imager on GOES-R, Bull. Am. Met. Soc., 86, 1079-1096, doi:10.1175/BAMS-86-8-1079, 2005.

Reply: Thank you for your suggestion. The recommended references below has been incorporated into the text.

Schmit, T. J., Gunshor, M. M., Menzel, W. P., Gurka, J. J., Li, J., and Bachmeier, A. S.: Introducing the next-generation advanced baseline imager on GOES-R, Bull. Am. Met. Soc., 86, 1079-1096, doi:10.1175/BAMS-86-8-1079, 2005.

Schmit, T. J., Griffith, P., Gunshor, M. M., Daniels, J. M., Goodman, S. J., and Lebair, W. J.: A closer look at the ABI on the GOES-R series, Bull. Am. Met. Soc., 98(4), 681–698. https://doi.org/10.1175/BAMS-D-15-00230.1, 2017.

(5) l.132, 133: The expression "image acquisition time" is somewhat misleading! AIRS is scanning repeatedly in the across-track direction taking footprints one-by-one. The AIRS data are then arranged into granules of 6min, each.

Reply: Thank you for comments. We have made the following revisions:

"AIRS performs continuous across-track scanning, acquiring data footprints sequentially. The collected data are then organized into 6-minute granules."

(6) Please provide references for the AIRS instrument! For example:

Aumann, H. H., et al.: AIRS/AMSU/HSB on the Aqua mission: Design, science objective, data products, and processing systems, IEEE Trans. Geosci. Remote Sens., 41, 253-264, 2003.

Chahine, M. T., et al.: AIRS: Improving weather forecasting and providing new data on greenhouse gases, Bull. Am. Met. Soc., 87, 911-926, doi:10.1175/BAMS-87-7-911, 2006.

Reply: Thank you for your suggestion. The recommended references have been incorporated into the text.

(7) p.8: Please provide in Sect.2 also some information about the SABER instrument because also SABER data are used later in the manuscript.

Reply: Thank you for your suggestion. We added the following description to section 2.2 of the main text.

"Sounding of the Atmosphere using Broadband Emission Radiometry (SABER) is one of four instruments on NASA's Thermosphere Ionosphere Mesosphere Energetics Dynamics (TIMED) satellite (Russell et al., 1999), launched on December 7, 2001. TIMED focuses on exploring the energy properties and redistribution in the MLT region, providing data to define the basic states and thermal balance of this area. SABER is a 10-channel broadband limb-scanning infrared radiometer (1.27-17 μm). It measures kinetic temperature through $CO_2$ emissions (15 μm Local Thermodynamic Equilibrium (LTE) below 90 km; 4.3 μm non-LTE above 90 km) with ±2-5 K accuracy. Simultaneously observing $O_3$ (9.6 μm), OH (1.6-2.0 μm), and $O_2$ (1.27 μm) emissions, it quantifies radiative cooling (up to 150 K/day) and chemical heating (~8 K/day) in the MLT region with 2-4 km vertical resolution."

(8) l.217: In Fig.4, upper row, there are also indications of 630nm wave structures that are superimposed on the OH signature that is highlighted by the yellow square. These wave fronts are perpendicular to the OH wave fronts. Similar findings in Fig.5. You should comment on this. Do you think these patterns are from a different wave?

Reply: Thank you very much for your careful review. Yes, you are right. They are from a different wave. Their phase fronts are incoherent, and there are differences in the directions of propagation. We have marked these in Figs. 4 and 5 respectively.

The following comments have been incorporated into the main text.

"There are also observed curved wave structures (thermospheric CGW no. 2) (indicated by green arrows) whose wave fronts are perpendicular to those of the contaminating OH wave fronts."

[Figure]

**Figure 4.** All-sky 630.0 nm images (top panel) and OH images (bottom panel) were both projected onto an altitude of 87 km with an area of 1000 km × 1000 km. The northeastward-propagating CGW (marked with a yellow dashed box) shows contamination from OH airglow emission. Thermospheric CGWs propagating northwestward confirmed in 630.0 nm images (top panel). The phase fronts of the thermospheric CGW nos.1 (red lines) and 2 (green lines) are superimposed onto the OH images (bottom panel).

**630.0 nm images projected onto 250 km**

[Figure]

**Figure 5.** All-sky 630.0 nm images projected onto an area of 2000 km × 2000 km showing the thermospheric CGW nos.1 (indicated by red arrows) and 2 (indicated by green arrows) at approximately 4 min intervals in the SMS station on 18 September 2023. The red dots mark the estimated centers of the thermospheric CGW. The northeastward-propagating CGW (marked with a yellow dashed box) exhibits artifacts influenced by OH airglow emission.

(9) About Fig.4: The OH images and OI images were taken at almost the same time for demonstrating the contamination effect. Later in the manuscript you determine the time that the CGW takes to propagate from the OH altitude to the OI altitude to be around 1 hour. Therefore you should mention that some of the mismatches in the wave patterns shown in Fig.4 might be related to this.

Reply: Thank you for your good suggestion. The following description has been added to the main text.

"As discussed above, the OH images and OI images were captured nearly simultaneously to illustrate the contamination effect in Fig. 4. Some of the wave pattern mismatches in Fig. 4 are due to the propagation time required for CGWs to travel from the OH altitude to the OI altitude."

(10) Fig.7: Suggest to replace the red text "large scale CGW" in the figure with just "large scale GW" because it is difficult to tell whether this would be part of a concentric GW pattern. Even in the text you do not use the expression "CGW" for this wave pattern.

Reply: Thank you for your suggestion. Fig. 7 has been modified as shown below:

[Figure]

**Figure 7.** Suomi-NPP satellite Day Night Band radiance observations of CGWs at 03:59:54 UT on 18 September 2023. Red triangle represents the SMS station, and the red dot represents the position of the fitted center of the CGW.

In the text, the corresponding "CGW" has been revised to "GW".

(11) Caption of Fig.11: Please state whether these images are from OH, or from OI.

 Reply: We have made the following modifications to the caption of Fig. 11.

"Figure 11. Simultaneous observations of mesopause CGWs using OH channel ground-based all-sky airglow imager and TIMED/SABER satellite measurements. The red triangle marks the location of the SMS station. The instantaneous field of view of TIMED/SABER is 0.7 mrad by 10 mrad."

(12) l.342: Please check! The double-peak structure is seen mainly during the second overpass in the 07:18:23 UT profile, but not so much during the first overpass.

Reply: Thank you for your comment. The following discussion has been added to the text.

"There are weak double-peak structures during the first overpass at 00:24:10 UT and 00:28:15 UT. In contrast, the double-peak structure is more prominent during the second overpass in the 07:18:23 UT profile."

(13) l.368, 369: Please state that the flux is calculated for the altitude of the OH layer.

Reply: Thank you for your suggestion. We have made the following revisions:

"Figure 13 shows the calculated vertical flux of the horizontal momentum flux of mesopause CGWs in the altitude of the OH layer from 22:00 to 09:00 UT on 17-18 September 2023."

(14) l.374: How does this momentum flux compare to average values determined from SABER satellite data? A climatology is given, for example, in Ern et al. (2018).

Ern, M., Trinh, Q. T., Preusse, P., Gille, J. C., Mlynczak, M. G., Russell III, J. M., and Riese, M.: GRACILE: a comprehensive climatology of atmospheric gravity wave parameters based on satellite limb soundings, Earth Syst. Sci. Data, 10, 857-892, https://doi.org/10.5194/essd-10-857-2018, 2018.

Reply: Thank you for your suggestion. We compared our results with the gravity wave flux observations from satellite limb soundings by Ern et al. (2018), as described below.

"Ern et al. (2018) studied the climatology momentum flux determined from SABER satellite limb sounding data. They find that the GW absolute momentum flux is approximately 1–4 m²s⁻² in the mesopause region."

(15) l.387: The parameter alpha does not occur in Eq. (6), but only later in Eq. (7). Therefore the introduction of alpha should be moved there.

Reply: Thank you. The introduction of $\alpha$ has been moved after Eq. (7).

TECHNICAL COMMENTS:

l.18: CGWs -> concentric gravity waves (CGWs)

Reply: It has been revised.

l.81: its role -> their role

Reply: It has been revised.

l.250: Sumi -> Suomi

Reply: It has been revised.

l.301-307: Same sentence appears twice. Delete one of them.

Reply: It has been revised.

l.332: saber -> SABER

Reply: It has been revised.

l.356: are expressed -> is expressed

Reply: It has been revised.

l.358: is cancellation factor -> is a cancellation factor

Reply: It has been revised.

l.417: can be -> and can be

Reply: It has been revised.

l.475: for downloaded -> for download

Reply: It has been revised.

l.479: delete "radiances data" (double occurence).

Reply: It has been revised.

l.584: publication year of Heale et al. is 2022, not 2021.

Reply: It has been revised.

---

## Author Comment (AC2)

General comments

The study presents detailed observations of intense concentric gravity waves (CGWs) in the mesosphere and thermosphere over southern Brazil during 17–18 September 2023, triggered by fast-moving severe thunderstorms. Utilizing dual-channel ground-based airglow imaging (OH and OI 630.0 nm) alongside multi-satellite data (GOES-16, AIRS, VIIRS, SABER), the authors documented three CGW events lasting over 10 hours, with amplitudes exceeding 24% and horizontal movements over 400 km. The findings highlight exceptional momentum flux and vertical energy transport from the troposphere to the mesosphere–lower thermosphere (MLT) region. The study also addresses contamination in 630.0 nm thermospheric imaging due to OH emissions and explains the observed asymmetric wave propagation via Doppler effects from background winds. This work advances understanding of atmospheric coupling and underscores the value of coordinated multi-layer observations.

The study is scientifically sound and presents a comprehensive and well-supported analysis of extreme concentric gravity waves using an impressive combination of ground-based and satellite observations. The paper is well written, generally concise, and includes clear figures that support the findings. However, in a few instances, the inclusion of additional details, particularly regarding data interpretation and methodological assumptions, could enhance clarity and aid reader comprehension. I recommend accepting the paper for publication, subject to minor revisions.

We sincerely appreciate your time and effort in reviewing our manuscript, as well as your constructive feedback, which has greatly helped us improve the quality of our work. We have carefully addressed all your comments and revised the manuscript accordingly.

Specific comments

lines 33-34: Might be helpful to mention the typical height of the OH airglow layer (~87 km) and OI airglow layer (~250 km).

Reply: Thank you for your suggestion. The modifications we have implemented are as follows:

"The same CGW event was observed propagating from the OH airglow layer (~87 km) to the thermospheric OI 630.0 nm airglow layer (~250 km)."

lines 40-46: While the discussion provides useful context on the sources of atmospheric gravity waves (AGWs), it would benefit from the inclusion of some earlier and potentially more foundational references. Citing key historical studies on different atmospheric gravity wave types and generation mechanisms would help establish a more comprehensive background for the reader.

Reply: Thank you for your suggestion. The following references were added to the reference list.

Fovell, R., Durran, D., and Holton, J. R.: Numerical simulations of convectively generated stratospheric gravity waves, J. Atmos. Sci., 49, 1427-1442, https://doi.org/10.1175/15200469(1992)049<1427:NSOCGS>2.0.CO;21992.

Fritts, D. C.: Shear excitation of atmospheric gravity waves, J. Atmos. Sci., 39, 1936–1952, https://doi.org/10.1175/1520-0469(1982)039<1936:SEOAGW> 2.0.CO;2, 1982.

Fritts, D. C., and Nastrom, G. D.: Sources of Mesoscale Variability of Gravity Waves. Part II: Frontal, Convective, and Jet Stream Excitation, Journal of the Atmospheric Sciences 49, 111–127, https://doi.org/10.1175/1520-0469(1992)049 <0111:SOMVOG>2.0.CO;2, 1992.

Nastrom, G. D., and Fritts, D. C.: Sources of Mesoscale Variability of Gravity Waves. Part I: Topographic Excitation, Journal of the Atmospheric Sciences 49, 101–110, https://doi.org/10.1175/1520-0469(1992)049<0101:SOMVOG>2.0.CO;2, 1992.

Piani, C., Durran, D., Alexander, M. J., and Holton, J. R.: A Numerical Study of Three-Dimensional Gravity Waves Triggered by Deep Tropical Convection and Their Role in the Dynamics of the QBO, J. Atmos. Sci., 57, 3689-3702, https://doi.org/10.1175/1520-0469(2000)057%3C3689:ansotd%3E2.0.co;2, 2000.

Plougonven, R., and Zhang, F.: Internal gravity waves from atmospheric jets and fronts, Rev. Geophys., 52, 33-76, https://doi.org/10.1002/2012RG000419, 2014.

lines 68-73: The authors should briefly explain what is meant by "dual-layer airglow observations" to provide clearer context for readers who may not be familiar with this technique. Specifically, clarifying that it involves simultaneous observations of airglow emissions from the mesosphere and thermosphere (e.g., OH and OI 630.0 nm layers) would help highlight the significance of this method for studying vertical wave propagation and atmospheric coupling.

Reply: Thank you for your suggestion. We have provided a brief clarification regarding dual-layer airglow observations as follows:

"Although the observation of AGWs by airglow imagers has been widely documented in previous studies (Dalin et al., 2024; Nyassor et al., 2021, 2022; Suzuki et al., 2007; Vadas et al., 2012; Vargas et al., 2021; Wüst et al., 2019; Xu et al., 2015; Yue et al., 2009), dual-layer airglow observations, which involve observing airglow emissions from a hydroxyl radical (OH) layer (~87 km) in the mesosphere and an atomic oxygen emission layer at 630 nm (OI 630.0 nm) (~250 km) in the thermosphere, offer a unique opportunity to simultaneously investigate CGWs in both the mesosphere and thermosphere. This configuration enables

comprehensive studies of gravity wave vertical propagation and their role in vertical atmospheric coupling. However, due to past limitations in observational capabilities, simultaneous detection of CGWs across both the OH and OI 630.0 nm layers was rare."

lines 77-82: It would be helpful to clearly state where and when the observations were conducted to orient the reader. Additionally, the reported 24% amplitude is striking, providing context by specifying which previous studies or typical values this is being compared to would clarify its significance.

Reply: Thank you for your suggestion. We have made the following revision:

"In this study, we observed multiple strong CGW events using airglow measurements in southern Brazil on 17-18 September 2023, with a maximum amplitude reaching 24%, which is far higher than previously reported events with average amplitudes of 2-3% (Li et al., 2016; Tang et al., 2014; Suzuki et al., 2007a). Through ground-based dual-layer and multi-satellite joint observations, we conducted a comprehensive analysis of these events to reveal their role in vertical energy transfer and atmospheric coupling."

lines 104-116: The authors should clarify what is actually done in step #2 of the image processing chain. Specifically, more detail is needed on how the van Rhijn effect and atmospheric extinction are corrected, what parameters are used, and how the corrections are applied to the data. This would help readers better understand the methodology.

Reply: Thank you for your suggestion. We provide a detailed description as follows:

Second, we corrected for the van Rhijn effect and atmospheric extinction using the approach described in Kubota et al. (2001). The observed airglow intensity $I(\theta)$ from the ground is not uniform across different zenith angles. This non-uniformity is due to the van Rhijn effect. Additionally, the observed airglow intensity is influenced by atmospheric extinction, which results from absorption and scattering along the line of sight.

Since airglow observations are subject to the van Rhijn effect, the measured emission intensity at a specific zenith angle (θ) follows the relation (Kubota et al., 2001):

$$I(\theta) = I(0) \cdot V(H, \theta),$$

$$V(H, \theta) = \left[ 1 - \left( \frac{R}{R+H} \right)^2 \sin^2(\theta) \right]^{-\frac{1}{2}}, \tag{1}$$

where $I(0)$ is the emission intensity at zenith. $V(H, \theta)$ is the van Rhijn correction factor. $R$ is the earth radius and $H$ is the height of OH airglow layer. The relationship

between the observed emission intensity $I(\theta)$—affected by atmospheric extinction—and the true emission intensity $I_{true}(\theta)$ at the airglow layer is described by Kubota et al. (2001).

$$I(\theta) = I_{true}(\theta) \cdot 10^{-0.4 \cdot a \cdot F(\theta)},$$

$$F(\theta) = [\cos\theta + 0.15 \cdot (93.885 - \theta \cdot \frac{180}{\pi})^{-1.253}]^{-1}, \tag{2}$$

where a is the atmospheric extinction coefficient, $F(\theta)$ is an empirical equation.

Consequently, the image correction factor, obtained from the combination of Eqs. (1) and (2), takes the form:

$$K = V(H, \theta) \cdot 10^{-0.4 \cdot a \cdot F(\theta)}. \tag{3}$$

The parameter $a$ depends on the atmospheric observing conditions. For the observed CGW events, we treat $a$ as temporally constant. By averaging the images over the observation period, we derive the zenith-angle-dependent airglow intensity profile. The optimal value of $a$ is determined by matching this observed profile with theoretical $K$ profiles across varying $a$. The fitted value of parameter $a$ is approximately 0.42. Finally, we apply the flat-field correction by dividing the raw images by the corresponding $K$ factor.

line 120: It seems the subsection introducing the SABER/TIMED measurements is missing?

Reply: Thank you for your suggestion. We added the following description to section 2.2 of the main text.

"Sounding of the Atmosphere using Broadband Emission Radiometry (SABER) is one of four instruments on NASA's Thermosphere Ionosphere Mesosphere Energetics Dynamics (TIMED) satellite (Russell et al., 1999), launched on December 7, 2001. TIMED focuses on exploring the energy properties and redistribution in the MLT region, providing data to define the basic states and thermal balance of this area. SABER is a 10-channel broadband limb-scanning infrared radiometer (1.27-17 μm). It measures kinetic temperature through $CO_2$ emissions (15 μm Local Thermodynamic Equilibrium (LTE) below 90 km; 4.3 μm non-LTE above 90 km) with ±2-5 K accuracy. Simultaneously observing $O_3$ (9.6 μm), OH (1.6-2.0 μm), and $O_2$ (1.27 μm) emissions, it quantifies radiative cooling (up to 150 K/day) and chemical heating (~8 K/day) in the MLT region with 2-4 km vertical resolution."

lines 122-129: The authors are kindly requested to provide a reference for the ABI (Advanced Baseline Imager) instrument onboard GOES-16 to support the description of its capabilities

Schmit, T. J., M. M. Gunshor, W. P. Menzel, J. J. Gurka, J. Li, and A. S. Bachmeier, 2005: INTRODUCING THE NEXT-GENERATION ADVANCED BASELINE IMAGER ON GOES-R. Bull. Amer. Meteor. Soc., 86, 1079–1096, https://doi.org/10.1175/BAMS-86-8-1079.

Reply: Thank you for your suggestion. The recommended reference has been incorporated into the text.

lines 133-135: The swath width of AIRS is approximately 1765 km, not 1600 km as stated (Hoffmann et al., 2014). I recommend citing Hoffmann et al. (2014) here, as their study offers important additional details on data processing methods—such as detrending—and discusses the sensitivity of AIRS stratospheric gravity wave observations, which are currently missing in this manuscript.

Hoffmann, L., Alexander, M. J., Clerbaux, C., Grimsdell, A. W., Meyer, C. I., Rößler, T., and Tournier, B.: Intercomparison of stratospheric gravity wave observations with AIRS and IASI, Atmos. Meas. Tech., 7, 4517–4537, https://doi.org/10.5194/amt-7-4517-2014, 2014.

Reply: Thank you for your suggestion. The recommended reference has been incorporated into the text.

lines 255-257: The relatively weak brightness temperature fluctuations observed by AIRS may result from the instrument's limited sensitivity to short vertical wavelengths (see, e.g., Hoffmann et al., 2024). Consequently, the observed brightness temperature amplitudes are typically much lower than the actual stratospheric temperature fluctuations, especially for convective wave events with short vertical wavelengths.

Reply: Thank you very much for your constructive suggestions. Your suggestions have been incorporated into the main text as follows:

"The relatively weak brightness temperature fluctuations observed by AIRS may result from the instrument's limited sensitivity to short vertical wavelengths (Hoffmann et al., 2014). Consequently, the observed brightness temperature amplitudes are typically much lower than the actual stratospheric temperature fluctuations, especially for convective wave events with short vertical wavelengths."

line 260: In Figure 6, the convective gravity waves (CGWs) might become more visible if the colorbar range is adjusted, for example, by using a fixed, symmetric range of ±0.5 K. Additionally, the colorbar label should be corrected to read "Brightness temperature perturbation (K)" instead of "Temperature perturbation (k)" to avoid confusion between measured radiance (brightness temperature) and actual atmospheric temperature.

Reply: Based on your suggestions, we have revised Figure 6 as shown in the figure below.

[Figure]

**Figure 6.** Aqua satellite 4.3 μm brightness temperature observations of CGWs at 05:05:21 UT on 18 September 2023. Brightness temperature is derived from 4.3 μm radiance at an altitude range of 30–40 km. The red triangle and dot mark the SMS station and fitted wave center, respectively.

lines 376-378: The statement "These events represent the most intense vertical transport cases ever recorded" should be better contextualized. Please clarify the criteria or dataset scope that support this claim to avoid potential overgeneralization.

Reply: Thank you very much for your comment.

We have removed the phrase "These events represent the most intense vertical transport cases ever recorded" from text to avoid potential overgeneralization.

lines 388-393: Another relevant study for comparison is Yue et al. (2013), which also presents multi-layer observations of convective gravity waves and estimates propagation times from the troposphere to the airglow layer, similar to the approach in this study. Including a discussion of their findings could provide valuable context and strengthen the interpretation.

Yue, J., L. Hoffmann, and M. Joan Alexander (2013), Simultaneous observations of convective gravity waves from a ground-based airglow imager and the AIRS satellite experiment, J. Geophys. Res. Atmos., 118, 3178–3191, doi:10.1002/jgrd.50341.

Reply: Thank you very much for your comment. The following discussion has been incorporated into the main text.

"Yue et al. (2013) conducted multilayer observations of convective gravity waves over the western Great Plains of North America and estimated that the time from the

convective source to the airglow layer was ~45 min."

lines 422-424: The authors should please clarify the actual detection threshold of the vertically integrated airglow observations, specifically the limit in terms of vertical wavelength.

Reply: Thank you very much for your comment. The following discussion has been incorporated into the main text.

"This strong eastward wind likely suppresses the visibility of eastward-propagating thermospheric CGWs in airglow imaging. We use Eq. 5 to estimate that the vertical wavelength of thermospheric CGWs propagating in the northwest direction is approximately 236 km, while that of thermospheric CGWs propagating eastward is approximately 62 km. The Doppler shift reduces their vertical wavelengths, causing them to fall below the detection threshold of the vertically integrated airglow observations, which is approximately 100 km from 200 km to 300 km during nighttime (Chiang et al., 2018)."

Chiang, C.-Y., Tam, S. W.-Y., and Chang, T.-F.: Variations of the 630.0 nm airglow emission with meridional neutral wind and neutral temperature around midnight, Ann. Geophys., 36, 1471–1481, https://doi.org/10.5194/angeo-36-1471-2018, 2018.

Technical corrections

line 18: The acronym "CGW" (Concentric Gravity Wave) should be introduced in full when first mentioned.

Reply: It has been revised.

line 28: change to "fast-moving deep convection" (singular)

Reply: It has been revised.

lines 304-307: Remove redundant sentence "Figure 9c present…"

Reply: It has been revised.

line 312: replace "ERA-5" by "ERA5"

Reply: It has been revised.

line 332: replace "saber" by "SABER"

Reply: It has been revised.

line 358: is _the_ cancellation factor

Reply: It has been revised.

lines 386-387: from _the_ troposphere to _the_ airglow layer

Reply: It has been revised.

---

## Author Comment (AC3)

General comments
The paper describes a case study of concentric gravity waves (CGWs) observed with the airglow imager in Brazil on 17-18 September 2023. Also, these CGWs were simultaneously captured by three satellites. Three groups of intense CGWs lasted over 10 hours. The CGWs caused profound airglow emission perturbations exceeding 24%. These CGW events were caused by fast-moving deep convections observed by the GOES-16 satellite. The authors have found that these CGW events represent the most intense vertical transport cases ever recorded, demonstrating remarkable wave coupling between the lower and upper atmosphere. I have found the paper to be interesting to the atmospheric community. At the same time, I have found a number of issues that should be explained in more detail. That is why I recommend accepting the paper after major revision.

We sincerely appreciate your thoughtful and constructive feedback on our manuscript. We are grateful for the time and effort dedicated to evaluating our work, especially the recognition of its significance to the atmospheric community. Your insightful comments have helped us identify areas where the study can be further strengthened. The detailed technical comments have helped us significantly improve the rigor and clarity of the study. Below, we address each point in detail and have incorporated substantial revisions to improve clarity, methodology, and discussion as suggested.

Specific comments
Line 71: "…dual-layer airglow observations…" It is not clear what dual-layer the authors talk about? It should be clarified here.

Reply: Thank you very much for your comment. We have provided the following clarification regarding dual-layer airglow observations:

"Although the observation of AGWs by airglow imagers has been widely documented in previous studies (Dalin et al., 2024; Nyassor et al., 2021, 2022; Suzuki et al., 2007a; Vadas et al., 2012; Vargas et al., 2021; Wüst et al., 2019; Xu et al., 2015; Yue et al., 2009), dual-layer airglow observations, which involve observing airglow emissions from a hydroxyl radical (OH) layer (~87 km) in the mesosphere and an atomic oxygen emission layer at 630 nm (OI 630.0 nm) (~250 km) in the thermosphere, offer a unique opportunity to simultaneously investigate CGWs in both the mesosphere and thermosphere. This configuration enables comprehensive studies of gravity wave vertical propagation and their role in vertical atmospheric coupling. However, due to past limitations in observational capabilities, simultaneous detection of CGWs across both the OH and OI 630.0 nm layers was rare."

Line 76: "… across these two atmospheric layers was rare." Across which two layers?

Reply: The two layers are OH and OI 630.0 nm layers, respectively.

Lines 86-87: "…São Martinho da Serra…" Please add Brazil here.

Reply: Thank you. The revision has been made as requested.

Lines 93-95: "The time resolution of the OH airglow image is 112 seconds, while that of the OI 630 nm airglow image is 225 seconds." Is it the time resolution or exposure time? What is the exposure time?

Reply: The time mentioned is the temporal resolution of airglow images. The exposure times of the OH airglow image and the OI 630 nm airglow image are 20 s and 90 s, respectively.

Line 98: "…the effective observation ranges of OH airglow imager with a 164°field of view" Before it was said that "fish-eye lens of a 180° field of view" What is true?

Reply: We appreciate your careful attention to the field of view (FOV) details. The fisheye lens itself has a 180° FOV. However, as you noted, we intentionally selected a 164° FOV as the effective observation range for our analysis. This is because the image distortion and stretching become increasingly severe near the edges of the fisheye lens, which would compromise the accuracy of our airglow measurements.

Lines 101-102: "Before effectively extracting the wave parameters, the raw airglow images need to be processed through the following steps:…" It is not a complete information on the processing of raw images. Among others, the following steps should be described: How the atmospheric background was subtracted? How the dark noise of the sensor was subtracted? How the flat field correction (non-uniformity of the sensor at different wavelengths) was taken into account? Was the imager absolutely calibrated in a lab? Does the imager register airglow intensities in relative or absolute units (Rayleigh)? At which solar depression angles does the imager operate?

Reply: My sincere apologies for not providing a detailed description of the process, which may have caused you confusion. We have provided the detailed process as follows:

"Before effectively extracting the wave parameters, the raw airglow images need to be processed through the following steps: First, a median filter with a kernel size of 17 × 17 pixels was employed to eliminate stars from the raw images (Li et al., 2011). We also removed the CCD dark noise, which was estimated from dark-frame images captured with the shutter closed prior to observations. Second, we corrected for the van Rhijn effect and atmospheric extinction using the approach described in Kubota et al. (2001). The observed airglow intensity $I(\theta)$ from the ground is not uniform across different zenith angles. This non-uniformity is due to the van Rhijn effect. Additionally, the observed airglow intensity is influenced by atmospheric extinction, which results

from absorption and scattering along the line of sight.

Since airglow observations are subject to the van Rhijn effect, the measured emission intensity at a specific zenith angle ($\theta$) follows the relation (Kubota et al., 2001):

$$I(\theta) = I(0) \cdot V(H,\theta),$$

$$V(H,\theta) = \left[ 1 - \left( \frac{R}{R+H} \right)^2 \sin^2(\theta) \right]^{-\frac{1}{2}}, \tag{1}$$

where $I(0)$ is the emission intensity at zenith. $V(H,\theta)$ is the van Rhijn correction factor. $R$ is the earth radius and $H$ is the height of OH airglow layer. The relationship between the observed emission intensity $I(\theta)$—affected by atmospheric extinction— and the true emission intensity $I_{true}(\theta)$ at the airglow layer is described by Kubota et al. (2001).

$$I(\theta) = I_{true}(\theta) \cdot 10^{-0.4 \cdot a \cdot F(\theta)},$$

$$F(\theta) = [\cos\theta + 0.15 \cdot (93.885 - \theta \cdot \frac{180}{\pi})^{-1.253}]^{-1}, \tag{2}$$

where a is the atmospheric extinction coefficient, $F(\theta)$ is an empirical equation.

Consequently, the image correction factor, obtained from the combination of Eqs. (1) and (2), takes the form:

$$K = V(H,\theta) \cdot 10^{-0.4 \cdot a \cdot F(\theta)}. \tag{3}$$

The parameter a depends on the atmospheric observing conditions. For the observed CGW events, we treat a as temporally constant. By averaging the images over the observation period, we derive the zenith-angle-dependent airglow intensity profile. The optimal value of a is determined by matching this observed profile with theoretical $K$ profiles across varying a. The fitted value of parameter a is approximately 0.42. Finally, we apply the flat-field correction by dividing the raw images by the corresponding $K$ factor.

Third, we eliminated atmospheric background counts from the images. For background emission, Swenson and Mende (1994) used simultaneous Infrared measurements to demonstrate that the background contributes approximately 30% of the total OH airglow image signal. Similarly, Suzuki et al. (2007b) confirmed this ratio (~30%) through concurrent OH intensity observations with a Spectral Airglow Temperature Imager. In this study, we adopt the same assumption that background emissions account for ~30% of the total signal."

The imager was not absolutely calibrated in the lab. As a result, it only measures

airglow intensities in relative units, not in absolute units like Rayleigh. Airglow observations are conducted when the solar depression angle is less than −12° 。

Swenson, G., and Mende, S. B.: OH emission and gravity waves (including a breaking wave) in all-sky imagery from Bear Lake, UT, Geophys. Res. Lett., 21, 2239–2242, https://doi.org/10.1029/94GL02112, 1994.

Suzuki, S., Shiokawa, K., Otsuka, Y., Ogawa, T., Kubota, M., Tsutsumi, M., Nakamura, T., and Fritts, D. C.: Gravity wave momentum flux in the upper mesosphere derived from OH airglow imaging measurements, Earth Planets Space, 59, 421–428, https://doi.org/10.1186/BF03352703, 2007b.

Line 117: "Third, the processed images were projected onto geographic coordinates,…" This information is not enough. It should be described in more detail how the optical model of the imager optical system was determined and calculated? What were the reference points in order to calculate free parameters of the optical model? Stars or lab reference points? What are the errors of projected pixels in the image center and at the edge of FoV? What is the spatial resolution of the imager in the imager center and at the edge of FoV?

Reply: We have provided a detailed description as follows:

"Then, the original airglow images were spatially calibrated using stars as reference points. Each pixel location (i, j) in the original image was first mapped to a position (f, g) in a standardized coordinate system. Subsequently, the point (f, g) was transformed into geographic coordinates (x, y) using azimuth (az) and elevation (el) angles.

The conversion between original image coordinates (i, j) and standard coordinates (f, g) is defined by a linear transformation (Hapgood and Taylor, 1982):

$$\begin{bmatrix} f \\ g \end{bmatrix} = \begin{bmatrix} a_0 & a_1 & a_2 \\ b_0 & b_1 & b_2 \end{bmatrix} \begin{bmatrix} 1 \\ i \\ j \end{bmatrix}, \tag{1}$$

where the coefficients a and b are calculated by applying a least-squares fitting using the observed location of the stars in the original image and their locations in standard coordinate (Garcia et al., 1997):

$$\begin{bmatrix} a_0 & b_0 \\ a_1 & b_1 \\ a_2 & b_2 \end{bmatrix} = \begin{bmatrix} \mathbf{1}^T\mathbf{1} & \mathbf{1}^T\mathbf{i} & \mathbf{1}^T\mathbf{j} \\ \mathbf{1}^T\mathbf{i} & \mathbf{i}^T\mathbf{i} & \mathbf{i}^T\mathbf{j} \\ \mathbf{1}^T\mathbf{j} & \mathbf{i}^T\mathbf{j} & \mathbf{j}^T\mathbf{j} \end{bmatrix}^{-1} \begin{bmatrix} \mathbf{1}^T \\ \mathbf{i}^T \\ \mathbf{j}^T \end{bmatrix} \begin{bmatrix} \mathbf{f} & \mathbf{g} \end{bmatrix}, \tag{2}$$

where the column vectors **i** and **j** contain observed star locations in the original image, while **f** and **g** hold their computed normalized coordinates. The vector **1** is a constant-valued column vector with length matching these vectors.

Through a georeference procedure, the standard coordinate images were projected onto geographic coordinates, assuming peak emission heights of 87 km for the OH layer and 250 km for the OI 630.0 nm layer. The spatial resolution of the imager varies significantly zenith angle. For the OH channel, it is 0.53 km/pixel at the center of the image and degrades to 39.8 km/pixel at the edge of the image. For the 630 channel, the resolution is 1.53 km/pixel at the center of the image and decreases to 40.8 km/pixel at the edge of the image."

You're absolutely right. There are projection errors. Hapgood and Taylor (1982) pointed out that there is some uncertainty in the position of any airglow structure due to the error in measuring the elevation and azimuth of the stars. The error of the stars in elevation is 2', corresponding error in the range to the airglow is ~1 km at 10° elevation, increasing to ~2 km at 5° elevation. They also studied the uncertainty in the position of any airglow structure caused by atmospheric refraction and found that it is less than the uncertainty arising from the error in measuring elevation. The CGWs we observed were almost exclusively within the field of view above an elevation angle of 10°; thus, these errors are not considered in this study.

Garcia, F. J., Taylor, M. J., and Kelley, M. C.: Two-dimensional spectral analysis of mesospheric airglow image data, Appl. Optics, 36, 7374–7385, https://doi.org/10.1364/AO.36.007374, 1997.

Hapgood, M. and Taylor, M. J.: Analysis of airglow image data, Ann. Geophys., 38, 805–813, 1982.

Lines 122-129. It should be given a reference on the GOES-16 satellite and addressed visible and infrared parameters.

Reply: Thank you for your comments. The revision has been made as requested.

"The Geostationary Operational Environmental Satellite-16 (GOES-16) (Schmit et al., 2005), launched in November 2016, is part of the GOES-R Series. The Advanced Baseline Imager (ABI) is the primary instrument on GOES-16, providing high-resolution imagery in 16 spectral bands, including 2 visible channels (0.47 µm and 0.64 µm), 4 near-infrared channels (0.86 µm, 1.37 µm, 1.6 µm, and 2.2 µm), and 10 infrared channels (3.9–13.3 µm), with a temporal resolution of 10 min and a spatial resolution of 0.5–2 km (Schmit et al., 2017). The brightness temperature (BT), derived from 10.3 µm infrared images from channel 13, is used to study the convection activities during the CGW events."

Schmit, T. J., Gunshor, M. M., Menzel, W. P., Gurka, J. J., Li, J., and Bachmeier, A. S.: Introducing the next-generation advanced baseline imager on GOES-R, Bull. Am. Met. Soc., 86, 1079-1096, doi:10.1175/BAMS-86-8-1079, 2005.

Schmit, T. J., Griffith, P., Gunshor, M. M., Daniels, J. M., Goodman, S. J., and Lebai, W. J.: A Closer Look at the ABI on the GOES-R Series. Bulletin of the American Meteorological Society, 98(4), 681–698, https://doi.org/10.1175/bams-d-15-00230.1, 2017.

Lines 139-141: "In this study, the $CO_2$ radiance emission band with frequencies ranging between 2299.80 $cm^{-1}$ and 2422.85 $cm^{-1}$ is utilized to measure stratospheric air temperature perturbations." It should be given a reference on the Aqua satellite and CO2 emissions used in this paper.

Reply: Thank you very much for your suggestion. The following references were added to the text.

Parkinson, C. L.: Aqua: an Earth-Observing Satellite mission to examine water and other climate variables, IEEE Transactions on Geoscience and Remote Sensing, 41(2), 173-183, https://doi.org/10.1109/TGRS.2002.808319, 2003.

Rothman, L. S., Gordon, I. E., Babikov, Y., Barbe, A., Chris Benner, D., Bernath, P. F., Birk, M., Bizzocchi, L., Boudon, V., Brown, L. R., Campargue, A., Chance, K., Cohen, E. A., Coudert, L. H., Devi, V. M., Drouin, B. J., Fayt, A., Flaud, J.-M., Gamache, R. R., Harrison, J. J., Hartmann, J.-M., Hill, C., Hodges, J. T., Jacquemart, D., Jolly, A., Lamouroux, J., Le Roy, R. J., Li, G., Long, D. A., Lyulin, O. M., Mackie, C. J., Massie, S. T., Mikhailenko, S., Müller, H. S. P., Naumenko, O. V., Nikitin, A. V., Orphal, J., Perevalov, V., Perrin, A., Polovtseva, E. R., Richard, C., Smith, M. A. H., Starikova, E., Sung, K., Tashkun, S., Tennyson, J., Toon, G. C., Tyuterev, Vl. G., and Wagner, G.: The HITRAN2012 molecular spectroscopic database, J. Quant. Spectrosc. Radiat. Transfer, 130, 4–50, http://dx.doi.org/10.1016/ j.jqsrt.2013.07.002, 2013.

Lines 143-144: "The Visible Infrared Imaging Radiometer Suite (VIIRS) instrument, onboard the Suomi NPP satellite…" It should be given a reference on the Suomi NPP satellite.

Reply: Thank you very much for your suggestion. The following references were added to the text.

Lee, T. F., Nelson, S.C., Dills, P., Riishojgaard, L.P., Jones, A., Li, L., Miller, S., Flynn, L.E., Jedlovec, G., McCarty, W, Hoffman, C., and McWilliams, G.: NPOESS: Next-generation operational global Earth observations, Bull. Am. Meteorol. Soc., 91, 727–740, https://doi.org/10.1175/2009BAMS2953.1, 2010.

Lewis, J. M., Martin, D. W., Rabin, R. M. and Moosmüller, H.: Suomi: Pragmatic visionary, Bull. Am. Meteorol. Soc., 91, 559–577, https://doi.org/10.1175/ 2009BAMS2897.1, 2010.

Lines 166-167: "CGW no. 1 first appeared in the southeast direction of the station." Is it in the southeast or in the southwest direction of the station?

Reply: I'm sorry. I made a mistake. CGW no. 1 first appeared in the southwest direction of the station.

Lines 172-175: "…the center moved approximately 436 km westward, with an average speed reaching ~65 km/h. This eastward drift of the wave's center could be indicative of the influence of prevailing wind patterns and the westward movement of the convective system itself." I hardly understand was it the eastward or westward drift? Or sometimes westward and sometimes eastward? This should be clarified.

Reply: I'm sorry. I made a mistake. The "westward" should be "eastward". We have made the following revisions:

"…the center moved approximately 436 km eastward, with an average speed reaching ~65 km/h. This eastward drift of the wave's center could be indicative of the influence of prevailing wind patterns and the eastward movement of the convective system itself."

Lines 173-174: "This eastward drift of the wave's center could be indicative of the influence of prevailing wind patterns…" This is very interesting but it is not entirely clear. At what altitude is the prevailing wind considered? In the tropopause or in the mesopause?

Reply: Thank you very much for your comments. The following description has been added to the main text.

"Apart from the moving convective system mentioned above, which is a primary cause of the eastward displacement of the CGW center observed at the mesopause, the prevailing winds near 10 km and 55 km in Fig. 9a also significantly contribute to the eastward movement of the CGW center."

Lines 176-177: "…are measured to be (30–82)±3 km." Having such a large range of wavelengths what is the physical sense of indicating the error of 3 km? What does this error tell us? Is it the instrumental error or geophysical wave variability or both? This is again connected to my above-mentioned questions on What are the errors of projected pixels in the image center and at the edge of FoV?

Reply: I highly admire your rigorous comments.

I'd like to clarify that the 30–82 km range indicates radial variations of horizontal wavelengths, while the ±3 km represents the variation of wavelengths along the circular arc direction.

In Figs. 2 and 3, what physical quantity can we see on these images? Is it some raw OH emission intensity? Or is it a corrected emission intensity? Or is it OH emission intensity in absolute values? It should be clarified in the figure captions.

Reply: Thank you very much for your comments. The following descriptions are added to captions of Figs. 2 and 3.

"The presented images display the corrected OH emission intensity."

In Fig. 3, it is difficult to see the green and light blue dots and arrows for color-blind readers. I recommend changing the green dots to, for example, green triangles.

Reply: Following your suggestions, we have made revisions to Fig. 3, as shown below:

[Figure]

**Figure 3.** All-sky OH images projected onto an area of 1000 km × 1000 km showing the CGW no. 2 and CGW no. 3 events at half-hour intervals in the SMS station on 18 September 2023. The red dot marks the estimated center of the CGW no. 1, while the green triangles and orange yellow dots indicate the estimated centers of the CGW no. 2 and CGW no. 3, respectively. The presented images display the corrected OH emission intensity.

Caption to Fig.6, please indicate the approximate altitude at which this temperature map is observed.

Reply: We have made the following modifications to the caption of Fig. 6.

"Figure 6. Aqua satellite 4.3 μm brightness temperature observations of CGWs at 05:05:21 UT on 18 September 2023. Brightness temperature is derived from 4.3 μm radiance at an altitude range of 30–40 km. The red triangle and dot mark the SMS station and fitted wave center, respectively."

Line 267: "…horizontal wavelengths are primarily distributed within the range of (38–52) ± 3 km" Having such a large range of wavelengths what is the physical sense of indicating the error of 3 km?

Reply: Thank you very much for your comments.

I'd like to clarify that the 38–52 km range indicates radial variations of horizontal wavelengths, while the ±3 km represents the variation of wavelengths along the circular arc direction.

Lines 301-307. The sentence is repeated twice, please remove the repeated part.

Reply: We have removed the repeated part.

Lines 306-307: "… when their phase velocities fall within the prohibited range." What is the prohibited range? How much is it?

Reply: We have elaborated on "prohibited range" as follows:

The dispersion relationship of GWs (Hines, 1960) is given by

$$m^2 = \frac{N^2}{(c-u)^2} - k^2 - \frac{1}{4H^2} ,\tag{1}$$

where $m$ and $k$ are vertical and horizontal wavenumbers, respectively; $c$ is the horizontal observed phase speed of the GW; $u$ is the background wind speed in the wave propagation direction; $N$ is the Brunt-Väisälä frequency; and $H$ is the scale height. According to Eq (1), if the GW propagation speed is close to or equal to the horizontal wind speed in the wave propagation, the vertical wave number $m$ will

become infinite, which means that when encountering a critical layer, GWs will not be able to propagate upwards.

A blocking diagram is used to represent the velocity distribution, and the GWs in this distribution area cannot propagate to a specific height due to the critical layer filtering effect. The blocking diagram is generated by the following equation:

$$c \leq V_z \cos \varphi + V_m \sin \varphi ,\tag{2}$$

here $V_z$ and $V_m$ are zonal and meidional wind speed, respectively. $\varphi$ is the azimuth (anticlockwise from the east) of the horizontal propagation direction.

"Prohibited range" refers to a speed range, as indicated by the areas at different times in Fig. 9c. The boundaries of these areas represent the maximum amplitude. In Fig. 9c, the maximum amplitude of the prohibited range is shown to be approximately 44 m/s.

[Figure]

**Figure 9.** (a) The (a) zonal and (b) meridional wind field profiles from ERA5 (0-70 km) and HWM14 model (70-87 km) at 21:00 UT, 23:00 UT, 01:00 UT, 03:00 UT, and 05:00 UT, respectively. (c) Two-dimensional blocking diagrams from 0 to 87 km derived from the wind profiles in (a) and (b) on 17-18 September 2023.

Lines 322-327: "We also conducted a statistical analysis of CGWs observed by a meridional airglow observation network across mainland China from September 2023 to August 2024, with data from selected stations including Daicai (25.34°N, 110.34°E), Wendeng (37.18°N, 121.79°E), Mohe (53.48N, 122.34°E), and Naqu (31.73°N, 92.47°E). The results indicate that the average CGW amplitudes ranged between 1.7% and 2.6%." It seems to me that this is a completely different study, with completely different regions than the area of Brazil under discussion. A reference to this study is needed here. Otherwise it should be removed.

Reply: Thank you very much for your suggestion. We have removed the sentences you mentioned above.

Lines 332-334: "During the generation and propagation of CGWs, two saber orbits passed over the station and happened to be within the field of view of the airglow imager, as shown in Fig. 11." How is the field of view of TIMED/SABER oriented in Fig.11? Please add this information.

Reply: Thank you very much for your comment. The field of view information has been added to the Fig.11 caption as shown below:

"Figure 11. Simultaneous observations of mesopause CGWs using OH channel ground-based all-sky airglow imager and TIMED/SABER satellite measurements. The red triangle marks the location of the SMS station. The instantaneous field of view of TIMED/SABER is 0.7 mrad by 10 mrad."

Lines 341-342: "In addition to this, we also observed a double-peaked structure in the airglow emission layer." Which airglow SABER profiles do demonstrate a double-peaked structure? This should be paid attention to.

Reply: Thank you for your comment. The following discussion has been added to the text.

"There are weak double-peak structures during the first overpass at 00:24:10 UT and 00:28:15 UT. In contrast, the double-peak structure is more prominent during the second overpass in the 07:18:23 UT profile."

Lines 342-346: "From the temperature profiles (Fig. 12b and d), we have detected a rich spectrum of vertically propagating waves with vertical wavelengths between 5 km and 20 km, which consists with concurrent airglow and satellite observations of upward-propagating CGWs." This sentence sounds very strange to me due to the following reasons: 1. There is no information at all about vertical spectrums of gravity waves derived from airglow and satellite observations. All presented data were about horizontal gravity wave patterns. Of course, using the dispersion relation for gravity waves one can derive a vertical wavelength from a horizontal wavelength, but it was not done in the manuscript so far. 2. Each presented temperature profile shows significant vertical variations, i.e., inside the FoV of the imager and outside it, far away from the imager. How can we be 100% sure that these temperature variations are due to CGWs and not other gravity waves? This sentence should be redeveloped or removed from the manuscript.

Reply: Thank you very much for your comments and suggestions, and we have removed the relevant description.

Equation 4. What is ω here and how was it calculated? What is g here?

Reply: I'm really sorry for missing the information of these two parameters.

$\omega = \dfrac{2\pi c_i}{\lambda_h}$ is the intrinsic frequency (where $c_i$ is the intrinsic phase speed), g is the

gravitational acceleration.

Lines 365-366: "…u is the wind speed in the wave direction derived from meteor radar,…" I could not find any information on a meteor radar used in this study. This information should be provided. Is a meteor radar located in the proximity to the imager? What is the accuracy of estimation of the horizontal wind speed from meteor radar data in the wave direction discussed here?

Reply: I sincerely apologize for the mistake. Initially, we intended to utilize wind field data from meteor radar, but the location was too far from the station. As a result, we ultimately used wind field data from HWM14.

Equation 6. Where is α in this equation? The authors do not provide information on how they estimated k, m, N parameters in relation to the vertical direction. I assume these parameters were calculated as mean values over the height range from the tropopause to the mesopause. But m and N may significantly vary with altitude, resulting in variations in the GW vertical group velocity (see for example, Fig. 4 in Dalin et al., 2016). This may provide significant deviation of the estimated propagation times. The author should provide a comment on Equation 6.

Reply: Thank you very much for your comments.

The definition of α should be introduced after Eq. (7) in the text. The following comments have been added to the main text.

"The horizontal wavenumber k is derived from airglow images. The Brunt-Väisälä frequency N and vertical wavenumber m were calculated as the mean value over the atmospheric layer spanning from the tropopause to the mesopause. Notably, the background wind and temperature may exhibit significant altitudinal variations, resulting in substantial variations in the CGW vertical group velocity."

Lines 388-390: "The vertical group velocities of CGW no. 1, CGW no. 2, and CGW no. 3 are estimated to be 31–37 ms−1, 24–30 ms−1, and 26–29 ms−1, respectively." What is the source of these estimated ranges of the vertical group velocities? The uncertainty of which parameters affects the uncertainty in the estimation of the group velocities to a greater extent?

Reply: Thank you very much for your comments.

The estimated ranges of the vertical group velocities are derived from CGW parameter measurements in airglow images as well as background atmospheric temperature and

wind fields. We have re-estimated the vertical group velocities of CGW no. 1, CGW no. 2, and CGW no. 3. The background temperature for calculating the vertical group velocity of CGW no. 1, no. 2, and no. 3 was derived from TIMED/SABER profiles within effective FOV of the OH imager during the first orbit, the average of the first and second orbits, and the second orbit, respectively, while wind field data combined ERA5 (0–70 km) and HWM14 (70–87 km).

The vertical group velocities of CGW no. 1, CGW no. 2, and CGW no. 3 are re-estimated to be 27–42 ms$^{-1}$, 21–32 ms$^{-1}$, and 24–31 ms$^{-1}$, respectively.

For CGW no.1, the uncertainty in the group velocity estimation mainly comes from the wave parameters derived from airglow images. For CGW no.2 and CGW no.3, variations in background temperature and wind fields contribute more to the uncertainty in vertical group velocities.

Figure 14. What are the red dashed lines in Fig.14 a and b? And I assume that the red triangle marks the location of the SMS station. Right?

 Reply: Thank you very much for your comments. We have made the following modifications to the caption of Fig. 14.

**"Figure 14.** (a) All-sky 630.0 nm imaging observation of thermospheric CGW (red dashed lines) at 01:41:57 UT on 18 September 2023. The yellow dot marks the estimated center of the thermospheric CGW. (b) All-sky OH imaging observation of mesospheric CGW at 00:54:48 UT on 18 September 2023. The red dashed lines mark out the mesospheric CGW with the same scale as the thermospheric CGW. The red dot marks the estimated center of the mesospheric CGW. (c) GOES-16 10.3 µm brightness temperature at 00:20:20 UT on 17-18 September 2023. The red triangle marks the location of the SMS station. (d) Wind profiles from ERA-5 (0-70 km) and HWM14 (70-250 km) averaged between 01:00 UT and 02:00 UT on 18 September 2023."

Lines 463-464: "…demonstrating their substantial impact on atmospheric dynamics and space weather." Please provide more information on how these waves substantially impact on space weather.

Reply: Thank you very much for your suggestions. We provide more information on how these waves substantially impact on space weather as below:

"… demonstrating their substantial impact on atmospheric dynamics and space weather by (1) seeding traveling ionospheric disturbances (TIDs) that disrupt communications/GPS, (2) triggering plasma instabilities, and (3) altering thermospheric density, affecting satellite drag."